# Comparison of six approaches to predicting droplet activation of surface active aerosol. Part 1: moderately surface active organics

Sampo Vepsäläinen[1], Silvia M. Calderón[1,2], Jussi Malila[1], and Nønne L. Prisle[1,3]

[1]Nano and Molecular Systems Research Unit, University of Oulu, P.O. Box 3000, FI-90014, Oulu, Finland
[2]Finnish Meteorological Institute, P.O. Box 1627, FI-70211, Kuopio, Finland
[3]Center for Atmospheric Research, University of Oulu, P.O. Box 4500, FI-90014, Oulu, Finland

**Correspondence:** Nønne L. Prisle (nonne.prisle@oulu.fi)

**Abstract.** Surface active compounds (surfactants) are frequently found in atmospheric aerosols and droplets. As they adsorb to the surfaces of microscopic systems, surfactants can decrease aqueous surface tension and simultaneously deplete the bulk concentration. These processes may influence the activation of aerosols into cloud droplets and investigation of their role in cloud microphysics has been ongoing for decades. In this work, we have used six different models documented in the literature to represent surface activity in Köhler calculations of cloud droplet activation for particles consisting of one of three moderately surface active organics (malonic, succinic or glutaric acid) mixed with ammonium sulphate in varying mass ratios. For each of these organic acids, we find that the models predict comparable activation properties at small organic mass fractions in the dry particles, despite large differences in the predicted degree of bulk-to-surface partitioning. However, differences between the model predictions for the same dry particles regarding both the critical droplet diameters and supersaturations increase with the organic fraction in the particles. Comparison with available experimental data shows that models assuming complete bulk-to-surface partitioning of the moderately surface active component (total depletion of the bulk) do not adequately represent the droplet activation of particles with high organic mass fractions. When reduced droplet surface tension is also considered, these predictions somewhat improve. Models that consider partial bulk-to-surface partitioning of surface active components yield results comparable to experimental supersaturation data, even at high organic mass fractions in the particles, but predictions of the degree of organic bulk–surface partitioning strongly differ. This work highlights the need for using a thermodynamically consistent model framework to treat surface activity of atmospheric aerosols and for firm experimental validation of model predictions across a wide range of droplet states relevant to the atmosphere.

## 1 Introduction

The effect of atmospheric aerosols on the climate is still among the largest uncertainties to estimates and interpretations of the Earth's changing energy budget (IPCC, 2013; Seinfeld et al., 2016). An aerosol population can be composed of dozens of inorganic salts mixed with hundreds of organic species. Single-particle measurements have shown mixtures of compounds

from primary sources, such as soot, dust and organic carbon, mixed with sulfate, nitrate and oxidized organics from secondary aerosol formation (e.g. Li and Shao, 2009; Gieré and Querol, 2010). Surface active organic species (surfactants) are frequently

found in atmospheric aerosols from many different regions and environments (e.g. Gérard et al., 2016; Petters and Petters, 2016; Nozière et al., 2017; Kroflič et al., 2018; Gérard et al., 2019). In liquid aerosol mixtures, such as aqueous droplets, surfactants can adsorb at the interfaces, lowering the surface tension and distributing their mass between the droplet bulk and surface phases. The distribution of surfactant mass between the surface and bulk of a solution is here referred to as the bulk–surface *partitioning*.

The various effects of surface activity for cloud microphysics have been investigated for decades, starting with Hänel (1976), who discussed the importance of possible surface tension deviations from the value for pure water on the equilibrium equivalent size of droplets, also noting how lowered surface tension values may lead to a decrease in critical supersaturation and an increase in critical droplet size. Surfactants could therefore potentially modify the cloud activation properties of aerosols by lowering the surface tension at the air–droplet interface. Shulman et al. (1996) presented model calculations demonstrating this

effect and showed that reduced surface tension of aqueous droplets can affect the shape of the Köhler growth curves (Köhler, 1936), lowering the required critical supersaturation for cloud droplet activation. Facchini et al. (1999) soon after measured the surface tension of atmospheric bulk samples, noting a significant surface tension depression at total organic concentrations considered to be representative of activating droplets, and further demonstrated that similar reductions in surface tension of activating cloud droplets could translate to a significant change in the cloud radiative forcing on a global scale. Facchini

et al. (2000) then confirmed that organic surfactants are indeed present in atmospheric aerosol, supporting the ability of cloud condensation nuclei (CCN) to reduce the surface tension of atmospheric droplets.

These works did not consider that surface adsorption and the resulting concentration gradient between surface and bulk phases can also lead to significant depletion of the bulk phase in microscopic and submicron droplets (i.e. with diameters in the micrometer range or smaller, or below 1 $\mu$m, respectively) typically involved in cloud droplet activation, due to their

large surface area-to-bulk volume ratios (e.g. Prisle et al., 2010; Bzdek et al., 2020; Lin et al., 2020). In microscopic droplet solutions, the bulk–surface partitioning can be strongly in favor of the surface, whereas the bulk of a macroscopic solution is essentially an infinite reservoir and concentrations are unaffected by surface adsorption (e.g. Prisle et al., 2010; Lin et al., 2018, 2020). The effect of surfactant bulk–surface partitioning was first considered by Li et al. (1998), who used a calculation scheme where droplet bulk depletion was taken into account in the droplet surface tension and therefore the shape of the Köhler

growth curve. Sorjamaa et al. (2004) then proposed that similar partitioning effects must be included also in the solute (Raoult) effect of the droplet vapor equilibrium.

Sorjamaa et al. (2004) presented thermodynamic predictions showing that partitioning of the surfactant from the droplet bulk to the surface can limit the amount of dissolved solute and therefore reduce hygroscopic water uptake. The studies of Li et al. (1998) and Sorjamaa et al. (2004) focused on droplets comprising a model surfactant sodium dodecyl sulfate and suggested

that the isolated consideration of surface tension depression in cloud droplet activation without accounting for the surfactant partitioning effect on bulk hygroscopicity can lead to exaggerated potential of cloud droplet nuclei (CCN) activation. This has later been demonstrated both experimentally and in thermodynamic model calculations for particles comprising a range

of surface active compounds and their mixtures with soluble components (Li et al., 1998; Sorjamaa et al., 2004; Prisle et al., 2008, 2010; Kristensen et al., 2014; Hansen et al., 2015; Petters and Petters, 2016; Lin et al., 2018; Forestieri et al., 2018; Prisle et al., 2019; Prisle, 2021).

Experimental evidence for the role of surface activity in activating droplets has yet to form a consistent picture. The predicted bulk–surface partitioning effects were only recently verified experimentally for finite-sized droplets (specifically of 7-9 $\mu$m radius, but more generally referring to microscopic particles and droplets with finite surface-area-to-bulk-volume ratios) suspended in the air (Bzdek et al., 2020). The analysis of experimentally determined CCN activity of surface active aerosols is complicated by the interdependent effects on surface tension and solute effects in finite systems (Prisle et al., 2010; Lin et al., 2020), which cannot readily be deconvolved in experiments with microscopic or submicron droplets. Many studies have focused on estimating the surface tension depression of activating droplets (e.g. Padró et al., 2010; Giordano et al., 2013). However, Prisle et al. (2008, 2010) found that experimentally observed CCN activation of straight chain fatty acid sodium salts was consistent with only very modest surface tension reduction in droplets, compared to pure water. These results imply that using pure water surface tension may give more realistic predictions of CCN activity for organic aerosol in large scale simulations than surface tension calculated without consideration of bulk–surface partitioning in droplets (Prisle et al., 2012b), but such an approximation may not be generally applicable (e.g. Nozière et al., 2014; Petters and Petters, 2016; Lowe et al., 2019).

Fatty acids and their salts are a major class of organic compounds identified in atmospheric aerosols (e.g. Mochida et al., 2002, 2007; Cheng et al., 2004; Li and Yu, 2005; Forestieri et al., 2018) and often are relatively strong surfactants (see e.g. Prisle et al. (2008, 2010) and references therein), but also surfactants of moderate strength are abundantly present in the atmosphere. The behaviour described above for the fatty acid salts may be a limiting case and not representative across the whole range of surfactant strength and other molecular properties found in the atmosphere. The partitioning behavior for surface active compounds of moderate strength may be more complex and dependent on droplet concentration and size (governing the surface area-to-bulk volume ratio). It has been shown that partitioning between droplet bulk and surface can affect activation properties of surface active aerosol. Enhancement of hygroscopicity has been reported from surface tension effects for organosulfate products (Hansen et al., 2015), secondary organic aerosol containing dicarboxylic acids (Ruehl et al., 2016), marine primary organics (Ovadnevaite et al., 2017), and pollen extracts (Prisle et al., 2019). Therefore, thermodynamically consistent bulk–surface partitioning models are needed to fully represent surfactant properties (strength) and mixing states in such droplet systems.

Several models have been developed to be used with Köhler theory to calculate bulk–surface partitioning of surface active components and the resulting effects in growing droplets, including the Gibbs surface approach by Sorjamaa et al. (2004), Prisle et al. (2008, 2010), the molecular monolayer surface model by Malila and Prisle (2018), the liquid–liquid phase separation (LLPS) approach with possible partial surface coverage by Ovadnevaite et al. (2017) and the compressed film surface model by Ruehl et al. (2016). Each of these models rely on a unique set of assumptions and requirements for application. In addition, simplified models, emulating the results of more comprehensive frameworks (Prisle et al., 2011), with further simplifying assumptions (Ovadnevaite et al., 2017), derived as analytical approximations to ease the computational load (Topping,

2010; Raatikainen and Laaksonen, 2011) have also all been employed. A more extensive overview of different bulk–surface partitioning approaches is given by Malila and Prisle (2018). Many of the models have been shown to agree well with experimentally observed CCN activity for selected model aerosol systems (e.g. Ruehl et al., 2016; Lin et al., 2018; Davies et al., 2019). A few studies have presented the results from different models for the same droplet systems. Lin et al. (2018) compared the models of Prisle et al. (2010), Prisle (2021) and Malila and Prisle (2018) for multiple droplet systems comprising of succinic acid, sodium dodecyl sulphated (SDS) and Nordic Aquatic Fulvic Acid (NAFA) mixed with sodium chloride as well as pollenkitts mixed with ammonium sulphate. Davies et al. (2019) used the model of Ruehl et al. (2016) and different variations of the models presented in Ovadnevaite et al. (2017) for a system of ammonium sulphate particles coated with suberic acid. The models each offer a different description of the phenomena related to surface partitioning and surface tension in small droplets. Each of the models have previously been compared with experimental data (e.g. Prisle et al., 2008, 2010; Ruehl et al., 2016; Lin et al., 2018; Davies et al., 2019) for a limited set of droplet systems and conditions. However, critical supersaturation ($SS_c$) data can only validate model predictions of the critical point of droplet activation and does not allow for a direct assessment of predictions of the bulk–surface partitioning in the droplets. The flexibility and robustness of the models, in terms of their ability to describe systems and conditions other than those used for their validation so far remains an open question. To our knowledge, the extent to which all the available models predict consistent droplet growth and activation properties for the same surface active aerosol systems and conditions has so far not been investigated. When models are applied in conditions for which they have not been directly validated, as for example across a broad range of aerosols and conditions in large scale simulations (Prisle et al., 2012b; Lowe et al., 2019), it is important to know whether significant differences in predicted droplet activation and cloud properties could occur, depending on the choice of surface activity model.

In this study, we have used the thermodynamic and simplified partitioning models of Prisle et al. (2010), Prisle et al. (2011), Malila and Prisle (2018), Ruehl et al. (2016), Ovadnevaite et al. (2017), as well as a general bulk solution model, all implemented into the same Köhler model framework. Each of the models are used to calculate Köhler curves for particles consisting of a surface active organic of moderate strength mixed with ammonium sulphate. Surface active organics of moderate strength are represented by dicarboxylic acids due to their atmospheric relevance (e.g. Shulman et al., 1996; Hori et al., 2003) and abundance (e.g. Khwaja, 1995; Mochida et al., 2007; Jung et al., 2010). The thermodynamic properties required for the simulations are relatively well constrained for dicarboxylic acids (e.g. Booth et al., 2009; Hyvärinen et al., 2006; Ruehl et al., 2016). We compare the Köhler growth curves predicted with the different models to investigate impacts of the significant differences between the model assumptions and the expected high sensitivity of the partitioning equilibrium for the moderately strong surfactant to the assumed droplet state. A separate companion study is currently in preparation, comparing predictions with the same models for particles comprising stronger surfactants.

## 2 Theory and modeling

Six different modeling approaches are used to estimate the possible surfactant effects in Köhler calculations of droplet growth and CCN activation. Descriptions of the different model calculations are presented in the following sections. Each of the bulk–

surface partitioning models were implemented into the same Köhler model framework running in MATLAB (2019, 2020). Models that were not previously developed by Prisle and co-workers, were built in MATLAB (2019, 2020) with the information provided in the presenting publications and by the authors in personal communication. Köhler growth curves were simulated for dry particles consisting of one of the dicarboxylic acids, malonic acid, succinic acid or glutaric acid, mixed with ammonium sulphate in mass fractions ranging from 0.2–0.95. In all cases, droplet activation is evaluated based on the maximum value of the calculated Köhler curve.

We describe surfactant strength similarly to Prisle et al. (2010, 2011) in terms of the ability to reduce the surface tension from that of pure water at a given surfactant bulk phase concentration. To reduce the surface tension of an aqueous bulk solution at 298.15 K by 10 % from the value of pure water, the mole fraction of malonic, succinic and glutaric acids in the solution must be 0.061, 0.017 and 0.0070 respectively (Hyvärinen et al., 2006). Furthermore, in sufficiently large concentrations the dicarboxylic acids studied here can at most reduce aqueous surface tension to roughly 50 mN m$^{-1}$ (Hyvärinen et al., 2006; Booth et al., 2009). Stronger surfactants, such as fatty acid salts, can reduce surface tension to 20–30 mN m$^{-1}$ and a given surface tension reduction occurs at much lower aqueous concentrations (Prisle et al., 2010).

## 2.1 Köhler theory

Cloud droplets form in the atmosphere when water vapor condenses on the surfaces of aerosol particles. The Köhler equation (Köhler, 1936) describes the process, relating the equilibrium water vapor saturation ratio ($S$) over a spherical solution droplet to its diameter ($d$) as

$$S \equiv \frac{p_w}{p_w^0} = a_w \exp\left(\frac{4\bar{v}_w \sigma}{RTd}\right), \tag{1}$$

where $p_w$ is the equilibrium partial pressure of water over the solution droplet, $p_w^0$ is the saturation vapor pressure over a flat surface of pure water, $a_w$ is the droplet solution water activity, $\bar{v}_w = M_w/\rho_w$ is the molar volume of water, $\sigma$ is the droplet surface tension, $R$ is the universal gas constant, and $T$ is the temperature in Kelvin. The water vapor supersaturation is generally defined as $SS = (S - 1) \cdot 100\%$ and droplet activation is determined in terms of the critical supersaturation ($SS_c$) or critical saturation ratio ($S_c$), corresponding to the maximum value of the Köhler curve described by Eq. (1). Köhler theory is used in all cases in this study to describe the formation of cloud droplets, but the treatment of bulk–surface partitioning of surface active species and the resulting droplet water activity and surface tension varies between the models.

All the Köhler calculations are initiated by defining a dry particle size and composition which determines the total amount of solute in the growing droplets. The calculations with all models assume spherical dry particles and additive solid phase volumes. For each dry particle size, the total amounts of ammonium sulphate and organic molecules are calculated based on their pure solid phase densities and relative mass fractions in the particle. For each droplet size along the Köhler curve for a given dry particle, the total amount of water in the droplet phase is estimated from the droplet size via model specific methods, listed in Table 1. Here, *iterative* calculation refers to determining the total amount of water ($n_w^T$) based on mass conservation using the ternary mixture density while *additive* calculation refers to the assumption of additive volumes of the dry particle and the condensed water (Hänel, 1976) and is calculated by subtracting the dry particle volume from the total droplet volume.

The equations for calculating the total amounts of salt, organic and water are presented in the section S2 of the supplement. The calculation of the amount of water in each droplet uses composition dependent density of the droplet solution (indicated in Table 1 as *iterative* calculations) when required by the model (bulk solution and monolayer models). For all other models (Gibbs, simple, compressed film and partial organic film models), additive volumes of water and the dry particle components were used as indicated in the original model descriptions.

The droplet solution is a ternary water–inorganic–organic mixture. As each droplet grows, the surface area-to-bulk volume decreases, which in turn affects the bulk–surface partitioning of surface active species (Prisle et al., 2010; Bzdek et al., 2020). The different partitioning models compared in this work make different assumptions regarding the partitioning of droplet components. While some consider only partitioning of the surface active species into a purely organic surface phase (Prisle et al., 2011; Ruehl et al., 2016; Ovadnevaite et al., 2017), others evaluate the full composition of both droplet surface and bulk phases (Prisle et al., 2010; Malila and Prisle, 2018). Fig. 1 offers a conceptual image of the initial conditions of the simulation represented by the dry particles of varying compositions, and of the different approaches used to model the growth of the droplet and predict cloud droplet activation.

Table 1 lists how the different models calculate the initial amount of water, the water activity, the surface tension of the droplets and what compounds reside in the droplet surface and bulk while the sections S2 and S4 of the supplement contain the details for the calculations for the amount of water in the droplets, the water activity and the fitted surface tensions used by the Gibbs, monolayer and bulk solution models. The water activity of the binary mixture of ammonium sulphate and water used with the simple model is calculated using a parametrization (Prisle, 2006) that has been previously used together with the model (e.g. Prisle et al., 2019; Prisle, 2021). Similarly to the original work describing the partial organic film model (Ovadnevaite et al., 2017), the activity used for this model is based on AIOMFAC calculations (Zuend et al., 2008, 2011; AIOMFAC-web). A significant challenge for Köhler calculations is that activity coefficients are not typically available for all droplet components and at solution states corresponding to growing and activating droplets. For some mixtures, activities can vary significantly across the relevant range of droplet compositions (e.g. Hyttinen et al., 2020; Michailoudi et al., 2020). Robust composition dependent activity relations are exceedingly difficult to obtain due to the challenges of related to their direct measurements for non-volatile and trace components and as activities cannot be inferred from the Gibbs–Duhem equation for higher order mixtures. To our knowledge, experimental data is not available to obtain a complete description of the non-ideal interactions in the ternary organic-inorganic aqueous mixtures relevant for the present work. Therefore, in the cases where the droplet bulk is a ternary mixture (Gibbs, monolayer, compressed film and bulk solution models), we calculate $a_w$ as a corrected molar fraction including non-ideal effects of ammonium sulphate on water (see Eq. (S5) of the supplement). Non-ideal effects can be included implicitly by using composition dependent experimental data in cases where such data is available. The monolayer model (Malila and Prisle, 2018) implicitly includes non-ideal solution interactions through composition dependent experimental solution surface tension and density. However, for the droplet compositions in the present work, densities of ternary solutions are calculated for pseudo–binary ideal mixtures of water-salt and organic (section S4.2 of the supplement) and therefore do not fully capture all non-ideal interactions. Prisle (2021) directly used a fully ternary experimental $a_w$ parameterization together with a Gibbs bulk–surface partitioning model,which is limited to the specific aqueous mixtures of NaCl and Nordic

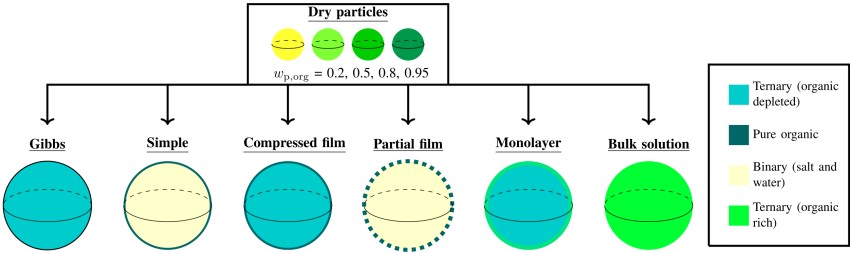

**Figure 1.** A conceptual figure of the different models. The figure shows the initial dry particles at a range of compositions (represented by the different colors of the particles), and the different models used to predict droplet critical properties. The different models are described at some point of droplet growth before activation, with the different colors of the droplet bulks or surfaces indicating differences in composition (dark turquoise for ternary mixture where the organic is significantly depleted, dark green for pure organic, pale yellow for binary mixture of water and salt, and bright green for ternary mixture rich in organic).

Aquatic Fulvic Acid studied. The compressed film model (Ruehl et al., 2016) parameters are fitted to experimental Köhler growth curve values and therefore includes an averaged account of the non-ideal solution interactions across the droplet states spanned by these experiments.

## 2.2 Gibbs adsorption partitioning model

In Gibbs surface thermodynamics, the gas–liquid interface is assumed to be an infinitely thin 2-dimensional surface called the Gibbs dividing surface, the location of which may vary depending on specific assumptions made for the system. The modeled idealized system is energetically and mechanically equivalent to the real system it represents (Defay and Prigogine, 1966). Several bulk–surface partitioning models for droplets have been developed based on Gibbs surface thermodynamics (e.g. Li et al., 1998; Sorjamaa et al., 2004; Prisle et al., 2010; Topping, 2010; Raatikainen and Laaksonen, 2011; Petters and Kreidenweis, 2013; Prisle, 2021; McGraw and Wang, 2021) but the assumptions related to the position of the dividing surface and the specific boundary conditions applied when solving the Gibbs adsorption equation (Gibbs, 1878) differ between the models.

In the model applied here (Prisle et al., 2010), the position of the Gibbs dividing surface is determined such that the bulk-phase volume ($V^B$) is equal to the total (equimolar) droplet volume ($V^T$) of all droplet components $j$. Compounds adsorbed at the surface are assumed to not contribute to the total droplet volume, and therefore a positive surface volume of one compound (surfactant) must be balanced by depletion of other compounds (water and salt) from the surface. Sorjamaa et al. (2004) combined the Gibbs adsorption equation with the Gibbs–Duhem equation for the droplet bulk, resulting in

$$\sum_j n_j^T kT \frac{d\ln\left(a_j^B\right)}{dn_{\text{org}}^B} + A \frac{d\sigma}{dn_{\text{org}}^B} = 0, \tag{2}$$

where $n_j^T$ is the total amount of species $j$ in the droplet solution, $k$ is the Boltzmann constant, $n_{\mathrm{org}}^B$ is the number of surfactant molecules in the droplet bulk, $a_j^B$ is the activity of $j$ in the droplet bulk solution, $A$ is the spherical droplet surface area and $\sigma$ is the droplet surface tension, given as function of bulk–phase composition. Equation (2) is solved iteratively for the bulk composition with the boundary condition that the molar ratio of water and salt is the same in the bulk and surface phases, such that the only adsorbing species is the surfactant. In addition, we assume volume additivity (such that the droplet diameter is given by the sum of individual pure component molar volumes) and mass conservation ($n_j^T = n_j^S + n_j^B$) of all components in the droplet (Prisle, 2006). The assumption of additive volumes is employed for calculating the total amount of water in the droplets at each droplet size along the Köhler growth curve. The calculation details relating to the total amount of water are given in the section S2.2 of the supplement.

## 2.3 Simple complete partitioning model

The simple model of Prisle et al. (2011) was developed to emulate the more complex Gibbs model of Prisle et al. (2010) specifically for predictions of $SS_c$. It approximates organic partitioning by simply assuming that all surface-active organics are partitioned to the droplet surface into an insoluble layer that is assumed to not affect the kinetics of water condensation–evaporation equilibrium. The surfactant solute therefore does not affect the water activity or surface tension of the aqueous droplet solution at the point of activation. This was shown to give very good representation of both the comprehensive model predictions and experimental results for droplet activation presented by Sorjamaa et al. (2004) and Prisle et al. (2008, 2010). The amount of surfactant in the droplet bulk is consequently vanishing ($n_{\mathrm{org}}^B = 0$), whereas neither salt nor water are present in the droplet surface. The total amount of water in the droplet is calculated assuming volume additivity of water and dry particle components. The surface tension of the droplet solution is assumed equal to that of pure water and therefore invariant with concentrations of both salt and surfactant in the droplet. Predictions with the comprehensive Gibbs partitioning model (Prisle et al., 2010) shows that this perhaps counter-intuitive condition is closely met in many droplet states, where the very large surface area-to-bulk volume ratios of microscopic droplets result in insufficient surface concentrations to significantly reduce surface tension, despite nearly all surface active material in the finite-sized droplets being partitioned to the surface (Prisle et al., 2010, 2011; Bzdek et al., 2020).

## 2.4 Compressed film surface model

The partitioning model of Ruehl et al. (2016) describes the surface as a compressed film (Jura and Harkins, 1946) and assumes phase separation in a droplet between the pure organic surface layer and droplet bulk. This framework is conceptually similar to the earlier van der Waals model of Ruehl and Wilson (2014), but applies different surface equations of state. Similarly to the Gibbs adsorption and simple complete partitioning models, only the organic partitioning is considered. The partitioning of surface active organic is assumed to take place into a 2-dimensional compressed film. As the droplet grows, the surface thickness decreases and eventually reaches a single monolayer, at which point the surface undergoes a phase transition to a non-interacting "gaseous" state and the surface tension reaches that of water (Forestieri et al., 2018). Typically, the compressed film model predicts activation to take place for droplets in this state (Ruehl et al., 2016).

Droplet growth and surfactant partitioning is treated at two different levels of iteration in the compressed film model. At the outer level, the equilibrium relative humidity (RH) is determined iteratively for each droplet diameter $d$ via Eq. (1). Water activity of the droplet bulk is calculated as the corrected mole fraction at each step of the iteration. The initially dry particles in the compressed film model are assumed to be composed of a salt core coated by a layer of organic, where both the diameters of the salt seed ($D_{\text{seed}}$) and coated particle ($D_p$) are known. The total amounts of all components in the droplet phase are

calculated with the assumption of volume additivity, both between the different components in the dry particle, and between the dry particle and water. The calculations use a different set of relations for the total amounts of salt, organic and water than the other partitioning models used in this work, in particular relating the salt and organic solid state molar volumes to the different diameters of the seed, coated and droplet diameters. For more information, see the Ruehl et al. (2016) supplement.

The inner level iteration takes place at the beginning of the outer level and calculates the fraction of organic molecules

adsorbed at the droplet surface, $f_{\text{surf}}$, using the isotherm for equation of state (EoS) of the compressed film model

$$\ln\left(\frac{C_{\text{bulk}}}{C_0}\right) = \frac{(A_0^2 - A^2)m_\sigma N_A}{2RT},$$  (3)

where $C_0$ is the bulk solution concentration at the 2-dimensional phase transition, $C_{\text{bulk}}$ is the bulk solution concentration, $A_0$ is the critical molecular area, $m_\sigma$ accounts for the interaction between surfactants at the interface and $N_A$ is Avogadro number. For each value of $f_{\text{surf}}$, values for $C_{\text{bulk}}$ and $A$ are calculated as

$$C_{\text{bulk}} = \frac{(1 - f_{\text{surf}})(D_p^3 - D_{\text{seed}}^3)\bar{v}_w}{d^3 \bar{v}_{\text{org}}}$$  (4)

and

$$A = \frac{6\bar{v}_{\text{org}}d^2}{f_{\text{surf}}(D_p^3 - D_{\text{seed}}^3)N_A}.$$  (5)

The droplet surface tension is calculated via an EoS that parameterizes $\sigma$ in terms of molecular area ($A$) and therefore relates $\sigma$ to the concentration of organics at the surface as

$$\sigma = \min\left(\sigma_w, \max\left(\sigma_w - (A_0 - A)m_\sigma, \sigma_{\min}\right)\right),$$  (6)

where $\sigma_{\min}$ is a lower limit imposed on the surface tension.

The model parameters $A_0, C_0, m_\sigma$ and $\sigma_{\min}$ are acquired through a separate fitting to experimental observations of ($d$, RH) using Eq. (1) in the range of droplet sizes before droplet activation (the rising part of the Köhler curve) at one organic fraction (Ruehl et al., 2016), or across several organic fractions at a fixed value of RH (Forestieri et al., 2018). The values used in

this work were obtained from Ruehl et al. (2016) and are provided in the supplement. These parameters are assumed to be compound specific physical constants and therefore to not be sensitive to seed or coated diameters or droplet dilution state, such that they can be applied across a range of organic fractions.

## 2.5    Partial organic film model

The partial organic film model used here is similar to the AIOMFAC-based simplified organic film model of Ovadnevaite et al.

(2017), where all organic is assumed to reside in an insoluble surface film adsorbed on an aqueous salt–rich bulk phase similar

to the assumption of Prisle et al. (2011) (permanent organic/inorganic phase separation). The film is assumed to not affect the kinetics of water condensation–evaporation equilibrium. The full AIOMFAC-based thermodynamic equilibrium model of Ovadnevaite et al. (2017) predicts possible liquid–liquid phase separation (LLPS), phase compositions and volumes, to describe droplets where partial surface coverage of a hygroscopic particle core ($\alpha$, bulk) by a organic rich phase ($\beta$, surface) is possible. In the model employed for this study, no water is present in the film, and it is assumed to coat the bulk entirely until a minimum surface thickness ($\delta_{\mathrm{org}}$) is reached and the organic film breaks, only partially covering the core phase for larger droplets. According to Ovadnevaite et al. (2017), the value roughly corresponds to an average molecular monolayer thickness, $\delta_{\mathrm{org}} = 0.16$ - $0.3$ nm, which is similar to the lengths scale of one to two covalent carbon-carbon bonds or van der Waals radii of carbon and oxygen atoms (Bondi, 1964). For the simulations in this study, $\delta_{\mathrm{org}}$ was set equal to the values given by the molecular surface monolayer model of Malila and Prisle (2018), which here predicts values between $0.39$ - $0.67$ nm, depending on the droplet size and the specific organic compound (figure in the section S1.3 of the supplement).

In the version of the model implemented here, water activity in the droplet bulk was calculated using a fit to AIOMFAC (Zuend et al., 2008, 2011; AIOMFAC-web) predictions as a function of salt mole fraction in the concentration range relevant for the growing droplets. The fit can be found in the section S2.3 of the supplement. The initial amounts of each compound in the droplets are determined via volume additivity, same as in the simple complete partitioning model of Prisle et al. (2011).

Following notation established in this study, we refer to the phase $\alpha$ as the bulk ($B$), and the phase $\beta$ as the surface ($S$). The model of Ovadnevaite et al. (2017) calculates the surface tension of an individual liquid phase as a volume fraction– weighted mean of the pure-component surface tension values ($\sigma_j$). For the bulk phase

$$\sigma^B = \sum_j \varphi_j^B \sigma_j, \tag{7}$$

where $\varphi_j^B$ is the volume fraction of component $j$ in the bulk phase ($\sum_j \varphi_j^B = 1$). The surface coverage parameter $c_S$ is defined as

$$c_S = \min\left(\frac{V^S}{V^\delta}, 1\right), \tag{8}$$

and determines whether the bulk is completely ($c_S = 1$) or partially ($c_S < 1$) covered by the organic film. Here, $V^S$ is the volume of the surface phase at diameter $d$ and $V^\delta$ is the corresponding volume of a spherical shell of thickness $\delta_{\mathrm{org}}$. The effective surface tension of the droplet is calculated as the surface area weighted mean of the surface tensions from both phases as

$$\sigma = (1 - c_S)\sigma^B + c_S \sigma^S. \tag{9}$$

### 2.6 Monolayer surface model

The molecular monolayer surface model of Malila and Prisle (2018) divides an aqueous droplet into a surface monolayer of thickness $\delta$ and droplet bulk with diameter $d - 2\delta$. The monolayer is described as a pseudo–liquid phase with a distinct composition from the droplet bulk and the monolayer composition is evaluated in terms of total amount of molecules for all

components in the droplet, not just the surfactant. For each species $j$, the partitioning between the bulk and surface phases is calculated iteratively from an extension of the Laaksonen–Kulmala equation (Laaksonen and Kulmala, 1991) relating the droplet surface tension $\sigma$ (as a function of bulk composition) to the surface composition as

$$\sigma(x^B, T) = \frac{\sum_j \sigma_j v_j x_j^S}{\sum_j v_j x_j^S}. \tag{10}$$

In Eq. (10), $v_j$ is the liquid (i.e. droplet surface) phase molecular volume, $\sigma_j$ the surface tension, and $x_j^S$ and $x_j^B$ the droplet surface and bulk mole fractions, respectively, all for component $j$. The condition of mass conservation ($n_j^T = n_j^S + n_j^B$) is imposed on the calculation for each compound $j$. The total amount of water is determined iteratively at each droplet diameter by assuming mass conservation of all compounds in the droplet, together with the ternary mixture density, which is here a function of the droplet composition. Details of calculating the total amount of water are given in section S2.2 of the supplement. The thickness of the surface molecular monolayer is calculated as

$$\delta = \left( \frac{\pi}{6} \sum_j v_j x_j^S \right)^{1/3}. \tag{11}$$

## 2.7 The bulk solution model

Simulations with the different partitioning models described are compared to a model representing the droplet as a bulk solution, neglecting effects of surfactant partitioning between the droplet bulk and surface. All surfactant mass is assumed to be evenly distributed in the bulk phase, which is equivalent in volume to the whole droplet (no separate surface phase). In the implementation used in this work, the amount of water in the droplet is solved iteratively from the ternary solution density, similarly to the monolayer model (Malila and Prisle, 2018). The droplet surface tension is evaluated from a fit to the ternary data reported by Booth et al. (2009). Details of the fit are provided in the section S4.1 of the supplement.

## 3 Results and discussion

In the following sections, we present and discuss results of the Köhler model simulations with the different bulk–surface partitioning models. Köhler curves were calculated for dry particles of $D_p = 50$ nm consisting of one of the organic acids malonic acid, succinic acid or glutaric acid, mixed with ammonium sulphate in organic mass fractions ($w_{p,org}$) of 0.2, 0.5, 0.8 and 0.95. Results are presented for the predicted equilibrium supersaturations ($SS$), surface tensions ($\sigma$) and organic partitioning factors ($n_{org}^S/n_{org}^T$) of growing droplets at 298.15 K. Results below focus on malonic acid–ammonium sulphate particles. Table 3 presents the droplet diameters, supersaturations and surface tensions at droplet activation predicted with the different models. The pure component properties used in the calculations are presented in Table 2. Corresponding results for simulations with succinic and glutaric acids as the organic component are presented in the section S1 of the supplement, together with the required compound properties. Sensitivity analysis in regards to various parameters and models are presented in the section S3 of the supplement.

**Table 1.** Methods of calculating the total amount of water ($n_w^T$) in the droplets, the droplet water activity ($a_w$) and surface tension ($\sigma$), and the composition of the droplet surface and bulk phases used with the different models. Details of the water activity and surface tension equations are given in the sections S2 and S4 of the supplement.

| Model | $n_w^T$ | $a_w$ | $\sigma$ | Surface comp | Bulk comp |
|---|---|---|---|---|---|
| Gibbs | Additive | Corrected mole fraction | Fit to data[a] | org, salt, water[b] | org, salt, water |
| Simple | Additive | Fit to data[c] | $\sigma_w$ | org | salt, water |
| Compressed film | Additive | Corrected mole fraction | Eq. (6) | org | salt, water, org $(S < S_c)$[d] |
| Partial organic film | Additive | AIOMFAC[e] | Eq. (9) | org | salt, water |
| Monolayer | Iterative | Corrected mole fraction | Fit to data[a] | org, salt, water | org, salt, water |
| Bulk solution | Iterative | Corrected mole fraction | Fit to data[a] | - | org, salt, water |

[a] Fit into the data of Booth et al. (2009) at 294.15 K. Fit details are in the section S4 of the supplement. [b] Salt and water depletion from the surface balance the organic partitioning. [c] Prisle (2006) [d] Typically all organic has partitioned to the surface at activation. [e] Fit into AIOMFAC-web calculations.

**Table 2.** The molar masses ($M$), liquid and solid densities ($\rho_L$ and $\rho_S$) and surface tensions ($\sigma$) of the different droplet components at 298.15 K, unless stated otherwise.

| Compound | $M$ (g mol$^{-1}$) | $\rho_L$ (kg m$^{-3}$) | $\rho_S$ (kg m$^{-3}$) | $\sigma$ (mN m$^{-1}$) |
|---|---|---|---|---|
| Water | 18.0153 | 997.05[a] | - | 71.97[b] |
| $(NH_4)_2SO_4$ | 132.1388 | 1548.93[c] | 1769.0 [d] | 99.66[e] |
| Malonic acid | 104.062 | 1529.13[f] | 1619.0[d] (283.15 K) | 48.24[g] |

[a] Pátek et al. (2009), [b] IAPWS (2014), [c] Extended from Tang and Munkelwitz (1994) [d] CRC Handbook (1988) [e] Fit into data from Hyvärinen et al. (2005) and Aumann et al. (2010) (details in the section S4 supplement), [f] Topping et al. (2016) [g] Hyvärinen et al. (2006)

## 3.1 Köhler curves and droplet activation

Figure 2 shows the Köhler curves, in terms of equilibrium supersaturation as a function of droplet diameter, predicted with each of the partitioning models for dry particles with $D_p = 50$ nm and organic mass fractions ($w_{p,org}$) of 0.2, 0.5, 0.8 and 0.95. The critical point ($d_c$, $SS_c$) is in each case determined as the maximum value of $SS$ along the Köhler curve, together with the corresponding droplet diameter. It is immediately clear from Fig. 2 that the different partitioning models lead to significantly different predictions of both $d_c$ and $SS_c$ for the same particle compositions. The divergence between results of the different models increases with mass fraction of surface active malonic acid in the particles, highlighting the strong dependency of results on the representation of malonic acid bulk–surface partitioning in the droplets. The Köhler curves predicted for particles containing succinic or glutaric acid (presented in section S1 of the supplement) both have similar tendencies to the malonic acid predictions. The disparities between the results of the different models also increase with the mass fraction of the organic acid.

### 3.1.1 Critical point

For particles with $w_{\mathrm{p,org}} = 0.2$ in Fig. 2(a), the Gibbs model predicts the lowest $SS_c$, followed by the partial organic film, bulk solution and monolayer models, while the compressed film model and the simple partitioning model predict the highest values of $SS_c$. The order is reversed for the critical diameters ($d_c$), where the Gibbs model predicts the largest value, and the compressed film model the smallest (Table 3). Predicted $SS_c$ for all the partitioning models fall within a supersaturation range of 0.1%, while the values for $d_c$ vary across a range of about 64 nm. These ranges are simple absolute differences between the largest and smallest calculated values for $d_c$ and $SS_c$. All the models predict slightly lower $SS_c$ values compared to the experimental $SS_c$ by Abbatt et al. (2005) in Fig. 2(a). However, the experimental line is at slightly larger particle size and organic mass fraction in the particle than the different model predictions. The larger particle size will decrease the $SS_c$ value while the larger organic mass fraction will increase it.

At $w_{\mathrm{p,org}} = 0.5$ in Fig. 2(b), the relative order of $SS_c$ values predicted with the different partitioning models remains the same as in Fig. 2(a). The discrepancies between the Gibbs and partial organic film model predictions decreases. The compressed film model predicts a distinctly different shape of the Köhler curve, with the droplet growing considerably less under supersaturated conditions, compared to the predictions of the other models. The range of predicted $SS_c$ values increases compared to Fig. 2(a) and now $\Delta SS_c = 0.19$ %, while the predicted $d_c$ values are all within a range of $\Delta d_c = 84$ nm. The experimental $SS_c$ in Fig. 2(b) by Abbatt et al. (2005) corresponds best to the compressed film model predictions, while the simple model predicts slightly higher $SS_c$ and the rest of the models predict slightly lower $SS_c$ values. However the experimental $SS_c$ is at slightly smaller particle size and higher organic mass fraction in the particles than the different model predictions. The experimental $SS_c$ marked in Fig. 2(b) is therefore expected to be larger than the corresponding value at the simulation conditions. Lin et al. (2018) reported the monolayer model to predict lower $SS_c$ values than the Gibbs model for mixed succinic acid–NaCl particles with the same dry particle size and organic mass fraction. In the current study, the situation is reversed, also for succinic acid (results presented in the section S1 of the supplement). This is an effect of mixing with ammonium sulphate instead of NaCl, which can affect both surface activity of the organic (Prisle et al., 2012a) in the Gibbs framework, as well as the density of the surface phase in the monolayer framework.

At higher organic fractions, $w_{\mathrm{p,org}} = 0.8$ in Fig. 2(c), the differences between the predictions with models where all organic is assumed to be partitioned to the surface along the entire growth curve (simple model and partial organic film model) and the models which account for evolving bulk–surface partitioning (monolayer model, Gibbs model and compressed film model) becomes much more apparent. The simple partitioning model predicts a significantly higher $SS_c$ than the other models, reflecting the strong absence of predicted solute effect in the droplets as the organic fraction increases. The partial organic film model predicts $SS_c$ comparable to the other models, except for the simple model. After the simple model, values of $d_c$ predicted with the partial film model are the smallest among the different models. The Köhler curve predicted with the partial organic film model also includes a local minimum, matching the point where the droplet surface film breaks and no longer fully encloses the bulk phase. The critical point of activation predicted with the partial film model occurs at a droplet size where the surface film is intact. This was not observed in previous studies by Ovadnevaite et al. (2017) for their particles comprising a surrogate

organic mixture and ammonium sulphate with $w_{p,org} = 0.5$ and $D_p = 175$ nm or $D_p = 41$ nm, and neither by Davies et al. (2019) for particles containing suberic acid and ammonium sulphate with $w_{p,org} = 0.49$ at $D_p = 100$ nm or $D_p = 40$ nm. The Köhler curves predicted with the bulk solution, Gibbs, and monolayer models group together with only small differences. The Köhler curve calculated with the compressed film model has a distinct shape, compared to the other models, but the $SS_c$ is similar to the other model predictions excluding the simple model. The compressed film model predicts the largest $d_c$ value 2(c). The range of predicted $SS_c$ increases to $\Delta SS_c = 0.5$ % and the predicted $d_c$ range also increases to $\Delta d_c = 199$ nm.

At $w_{p,org} = 0.95$ in Fig. 2(d), the results for supersaturation show similar trends as in Fig. 2(c). The most notable difference to Fig. 2(c) is that the partial organic film model predicts comparatively higher $SS_c$ in Fig.2(d) than it did in Fig. 2(c). The compressed film model again predicts the largest $d_c$ at a slightly lower $SS_c$ value than the bulk solution, Gibbs and monolayer models. In addition, a significant second local maximum is visible for the curve calculated with the compressed film model. This maximum is also present for the other surface active acids (results presented in the section S1 of the supplement) and we therefore made a sensitivity analysis of the model at this mass fraction (presented in the section S3.1 of the supplement). The sensitivity analysis shows that predictions of the critical point with the compressed film model are stable with respect to relatively large variations in the model parameters for particles containing malonic and glutaric acid. The critical point destabilises (moves to the second local maxima) at perturbations of 9-52 % depending on the model parameter in question. The predictions for succinic acid particles are more sensitive to variations in model input parameters, as the perturbations required to move the critical point are only between 3-6 % for the different model parameters. The increasing trend in the width of the ranges of predicted $SS_c$ and $d_c$ with organic mass fraction in the dry particle continues to $\Delta SS_c = 1.31$ % and $\Delta d_c = 282$ nm, respectively. In Fig. 2(d) we have included the two experimental values for the $SS_c$ of malonic acid particles. The values on the left in Fig. 2(d) at $D_p = 52$ from Giebl et al. (2002), and the values on the right at $D_p = 48$ nm from Rissman et al. (2007), who evaluated it from the results of Kumar et al. (2003). The monolayer, Gibbs, bulk and compressed film models all predict comparable $SS_c$ to the experimental values, while the partial organic film and the simple partitioning model predict considerably larger $SS_c$ values. Although the experimental data is obtained at slightly different conditions than the present simulations, comparison with the $SS_c$ prediction of the simple model indicates that the model predicts $SS_c$ to be too high because underlying assumptions of complete surface partitioning are no longer representative for the highest organic fractions in the particles. These findings for moderately surface active organics are in line with observations by Prisle et al. (2011) for strong surfactants such as fatty acid salts and SDS, that predictions with the simple model differ from both experiments and the detailed Gibbs model (Prisle et al., 2010) at high organic mass fractions.

### 3.1.2 Shape of droplet growth curves

Predictions with the compressed film model are distinct from the other partitioning models (see Table 3), in that $SS_c$ values do not always increase and $d_c$ values do not always decrease as a function of $w_{p,org}$. This can be observed for all particle systems in this work (see section S1 of the supplement for results of succinic and glutaric acid particles). The results of Davies et al. (2019) show a similar trend for the $SS_c$ predictions of the compressed film model. The Köhler curves predicted with the compressed film model and the simple partitioning model merge for higher organic mass fractions, after the droplets grow

beyond the activation point of the compressed film model Köhler curve (as evident in Figs. 2(c) and 2(d) and also suggested in Fig. 2(b)). This is an effect of the closely similar conditions predicted with the two models after the critical point of the Köhler curve. The compressed film model predicts all organic to have partitioned to the surface by the point of activation and therefore the surface tension is equal to that of pure water for droplets after this point. The only differences between models for droplets in this size range are due to the Raoult terms, which are also very similar (with differences on the order of $10^{-5}$).

In the partial organic film model, complete surface partitioning of the organic component is assumed along the whole Köhler curve, but the surface tension is calculated via Eq. (9). The partial organic film model predictions agree well with the monolayer, Gibbs and bulk solution models for particles with lower organic mass fractions, with the Köhler curve falling between those predicted with the Gibbs and the other two models in Figs. 2(a) and 2(b). In Fig. 2(c), the $SS_c$ values are still similar, but the partial organic film model predicts activation at a noticeably smaller droplet size than the other models. For malonic acid particles with $w_{p,org}$ of 0.8 and 0.95 (Figs. 2(c) and 2(d)), Köhler curves predicted with the partial organic film model nearly converge with those predicted with the monolayer, Gibbs and bulk solution models after the monolayer breaks, however, for succinic and glutaric acid particles (see section S1 of the supplement), Köhler curves predicted with the partial organic film model show larger differences from the other models.

Prisle et al. (2019) modeled the CCN activity of six pollenkitts in pure pollenkitt particles or mixed with 20% by mass of ammonium sulphate, using the bulk solution, Gibbs and simple partitioning models. They found that the bulk solution model generally predicted significantly lower $SS_c$ values compared to the Gibbs and simple partitioning models but that none of the three models were able to capture the measured pollenkitt CCN activity well over the full range of particle sizes studied. In Fig. 2 the situation is different, as we find the Gibbs model predicting slightly lower $SS_c$ values than the bulk solution model for $w_{p,org} = 0.2, 0.5$ and $0.8$ (Table 3). These differences reflect the surface activity of the organic component with pollenkitt being a more strongly surface active substance than malonic acid. While pollenkitt is a complex mixture of acidic and other organic compounds, the substance has significant ability to reduce aqueous solution surface tension. According to Prisle et al. (2019) both pollenkitts used in the study are able to reduce aqueous surface tension to values below $50\,\mathrm{mN\,m^{-1}}$ at concentrations of about $0.1\,\mathrm{g\,L^{-1}}$. Estimating the surface tension in an aqueous solution of malonic acid at same concentration according to the fit of Hyvärinen et al. (2006), we do not see any surface tension depression. The surface tension of pure supercooled malonic acid is estimated by Hyvärinen et al. (2006) as $48.24\,\mathrm{mN\,m^{-1}}$ at 298.15 K, which is lower than any surface tension reported by the same study for the aqueous solutions of the organic acids.

Lin et al. (2018) found that for succinic acid–NaCl particles, across $D_p = 50 - 150$ nm for mass fraction range of $w_{p,org} = 0 - 1$, the monolayer model predicts slightly lower critical supersaturation compared to the Gibbs model. In Table 3, the predicted $SS_c$ of the monolayer model is slightly larger than that of the Gibbs model for $w_{p,org} = 0.2, 0.5, 0.8$ of malonic acid. For succinic acid-ammonium sulphate particles in Table S2 of the supplement, the monolayer model predicts slightly higher $SS_c$ across all calculated compositions. As mentioned in relation to Fig. 2(b), this is due to the different salt present in the particles.

Davies et al. (2019) calculated Köhler curves for 100 nm particles over a range of compositions for particles comprising suberic acid and ammonium sulphate. In their calculations, the partial film model consistently predicted lower $SS_c$ than the

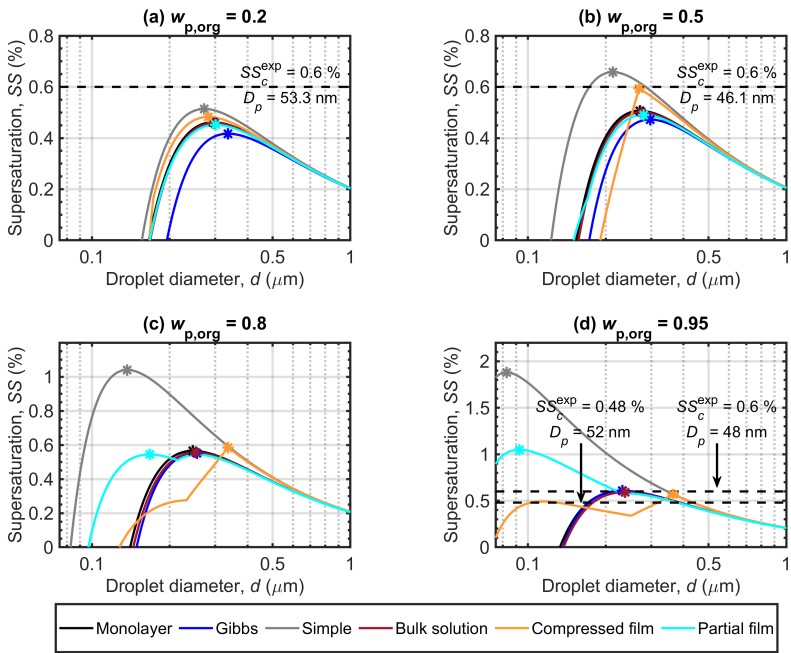

**Figure 2.** Köhler curves calculated with the different models for dry malonic acid–ammonium sulphate particles with $D_p = 50$ nm. Each panel shows curves for particles with a different malonic acid mass fraction ($w_{p,org}$). The critical points are marked on each curve. Experimental critical supersaturations reported by Abbatt et al. (2005) in panels (a) and (b) corresponding to $w_{p,org} = 0.3270$ and $0.5556$ respectively, and by Giebl et al. (2002) (values on the left corresponding to the lower dashed line) as well as by Rissman et al. (2007) (values on the right fitted to measurements by Kumar et al. (2003)) in panel (d) both for pure malonic acid particles (corresponding to $w_{p,org} = 1$), are also included. Note that the vertical axis scaling changes between the panels.

compressed film model for organic volume fractions up to about 0.8. This was also found for each of the organic acids in the present work, except at the highest mass fraction $w_{p,org} = 0.95$, which would correspond to a suberic acid volume fraction of 0.964 in particles mixed with ammonium sulphate.

### 3.2 Surface tension

Figure 3 shows the droplet surface tensions calculated with the different models along the Köhler curves in Fig. 2, for dry

particles of $D_p = 50$ nm and with malonic acid mass fractions ($w_{p,org}$) of 0.2, 0.5, 0.8 and 0.95 (shown in separate panels). The position of the critical point of droplet activation is indicated for each surface tension curve as $(d_c, \sigma_c)$, where $\sigma_c$ is the droplet surface tension evaluated at $d_c$. For each of the models, predicted surface tension can be significantly reduced at the smaller droplet sizes at the beginning of the growth curve where the surfactant is most concentrated in the droplets. However, in most cases, the surface tension at the point of activation is close to the pure water value. The simple partitioning model

yields a constant surface tension equal to that of pure water, as part of the basic assumptions for the model. The surface tension

of supercooled pure malonic acid is shown for reference, as a measure of the lowest physically realistic value for the droplet surface tension. The pure malonic acid surface tension is estimated according to the fit of Hyvärinen et al. (2006) (Table 2). The surface tension curves predicted for particles containing succinic and glutaric acids (presented in the section S1 of the supplement) show similar relative behavior to the curves predicted for malonic acid.

Figure 3(a) shows that for droplets formed on malonic acid particles with $w_{\mathrm{p,org}} = 0.2$, the surface tensions predicted along the Köhler curves with the monolayer, Gibbs and bulk solution models are similar, with the most visible differences at small droplet sizes. This indicates that the predicted droplet compositions are similar for all three models, as the droplet surface tension has been evaluated from the same composition-dependent function (given in Eq. (S9) of the supplement). The smallest droplets are both most concentrated and have the largest surface area-to-bulk volume ratios, so any differences in the representation of bulk–surface partitioning are expected to be more visible here. The droplet surface tensions predicted at activation ($\sigma_c$) are also very similar, as can be seen in Table 3. Quite to the contrary, the surface tension curve predicted with the compressed film model is very distinct. Droplet surface tension values start at a compound specific minimum surface tension value determined by the model parameter fitting (Ruehl et al., 2016), and then increase to the value for pure water towards the activation point. In Fig. 3(a), the first droplet size at which pure water surface tension is reached in the droplets does not correspond to the $d_c$ value (see Fig. 2, Table 3). This deviation from the typical behavior of the predictions as seen in Ruehl et al. (2016) could be an artefact since the model parameters are fitted to experimental observations for particles with high organic mass fraction and growing droplets in the range before the critical point of activation and are therefore not constrained by these measurements across all droplet states realized in our calculations. In the compressed film model, the fitted model parameters are assumed to be constant across varying organic mass fractions and dry particle sizes (concentrations), but for real droplet solutions mixing properties are likely sufficiently non-ideal (i.e. excess mixing properties are non-zero) that the model parameters would be expected to show some variation across the mixing space. Forestieri et al. (2018) made a similar observation for oleic acid particles within their parameter fitting range, at $w_{\mathrm{p,org}} = 0.8$ and NaCl seed particles of 80 nm. Forestieri et al. (2018) ascribed this behaviour to originate from the larger molecular volume of oleic acid, compared to the dicarboxylic acids investigated by Ruehl et al. (2016). In addition, the surface tension curves predicted with the compressed film model start at a minimum surface tension value below that of the measured surface tension of supercooled malonic acid. This is a result of the compressed film EoS presented by the original work of Ruehl et al. (2016) and given here in Eq. (6). The lower limit for the surface tension given by the equation is constrained by the model parameter $\sigma_{\mathrm{min}}$. The parameter is a fitted model parameter, which does not a priori have a physical interpretation and such may lead to unrealistic values of physical parameters as seen for the droplet systems investigated here. For the partial organic film model, droplet surface tensions start at the value of the pure organic compound (corresponding to complete surface coverage by the organic). The surface tension begins to increase once the organic film breaks as the droplet grows and the surface is no longer completely covered. In Fig. 3(a), all the $\sigma_c$ vales are within $3.1\ \mathrm{mN\,m^{-1}}$ of the surface tension of water (Table 3) and the lowest $\sigma_c$ is predicted with the partial organic film model.

The results for predicted surface tensions shown in Figs. 3(b), 3(c) and 3(d) for particles with higher mass fractions of malonic acid have very similar trends between the different models as seen in Fig. 3(a). The monolayer, Gibbs and bulk

solution models predict similar surface tensions and $\sigma_c$ decreases with increasing $w_{\mathrm{p,org}}$. With the compressed film model, the difference from Fig. 3(a) are the droplet sizes where the surface tension starts increasing from the minimum value and where activation occurs. In Figs. 3(b), 3(c) and 3(d), the activation point and the water surface tension are reached at the same droplet size. With the partial organic film model, the calculated droplet size at which the organic film breaks increases while $\sigma_c$ decreases with increasing $w_{\mathrm{p,org}}$, to eventually reach the pure organic surface tension value for the highest malonic acid mass fractions studied in Fig. 3(d). Of all the models compared here, both the partitioning models and the bulk solution model, the partial organic film model predicts the lowest surface tension at droplet activation, as can be seen in Fig. 3 and Table 3. Ovadnevaite et al. (2017) reported a reduced surface tension with simplified model calculations when compared to their full LLPS framework, indicating that the simplified model calculations provide a limit to the expected reduction in droplet surface tension during activation. However, Ovadnevaite et al. (2017) did not predict activation to occur before breaking of the organic film for roughly $w_{\mathrm{p,org}} = 0.5$ and at $D_p = 175$ nm or at $D_p = 41$ nm, as seen here in Figs. 3(c) and (d). Excluding predictions with the partial organic film model, the other predicted $\sigma_c$ values for malonic acid particle mixtures are within 1.3 mN m$^{-1}$ of the surface tension of water. This moderate surface tension depression agrees well with the predictions of Prisle et al. (2019) for pollenkitt modeled using the Gibbs model. However, Prisle et al. (2019) always predict considerably lower surface tensions (several tens of mN m$^{-1}$) at activation with the bulk solution model than with the Gibbs model. This is also the case here for all but the largest organic mass fraction, but the differences between the model predictions are considerably smaller ($\sim 0.1$ mN m$^{-1}$, Table 3).

## 3.3 Organic bulk–surface partitioning

Figure 4 shows the surface partitioning factors of malonic acid calculated with the different models for $D_p = 50$ nm particles with organic mass fractions ($w_{\mathrm{p,org}}$) of 0.2, 0.5, 0.8 and 0.95. The partitioning factor ($n_{\mathrm{org}}^S/n_{\mathrm{org}}^T$) is defined as the fraction of the total amount of organic molecules in the droplet which are predicted to reside in the surface. The simple partitioning model and the partial organic film model calculations are made with the assumption that all organic is always partitioned to the droplet surface and therefore the partitioning factor is equal to unity for all droplet states. The bulk solution model has no partitioning and therefore the partitioning factor is zero. Between the three models that calculate droplet state dependent partitioning (monolayer, Gibbs, and compressed film), large differences are seen in the predicted values of $n_{\mathrm{org}}^S/n_{\mathrm{org}}^T$ for malonic acid. At the point of activation, the compressed film model predicts nearly all malonic acid is partitioned to the droplet surface, whereas the Gibbs and monolayer models both predict a moderate fraction (well below 20%) of all malonic acid solute in the surface. These significant differences between the frameworks correspond to very different solution states for the same overall droplet compositions. Similar predictions were also observed for particles containing succinic and glutaric acids. More details can be found in section S1 of the supplement.

Figure 4(a) shows that for particles with malonic acid fractions of $w_{\mathrm{p,org}} = 0.2$, the monolayer model predicts stronger surface partitioning of malonic acid than the Gibbs model, especially at smaller droplet sizes. As the droplet grows, the partitioning factors predicted by the Gibbs model approach those of the monolayer model. Compared to the other two partitioning models, $n_{\mathrm{org}}^S/n_{\mathrm{org}}^T$ predicted by the compressed film model is very pronounced, always above 0.9. We also note a decrease

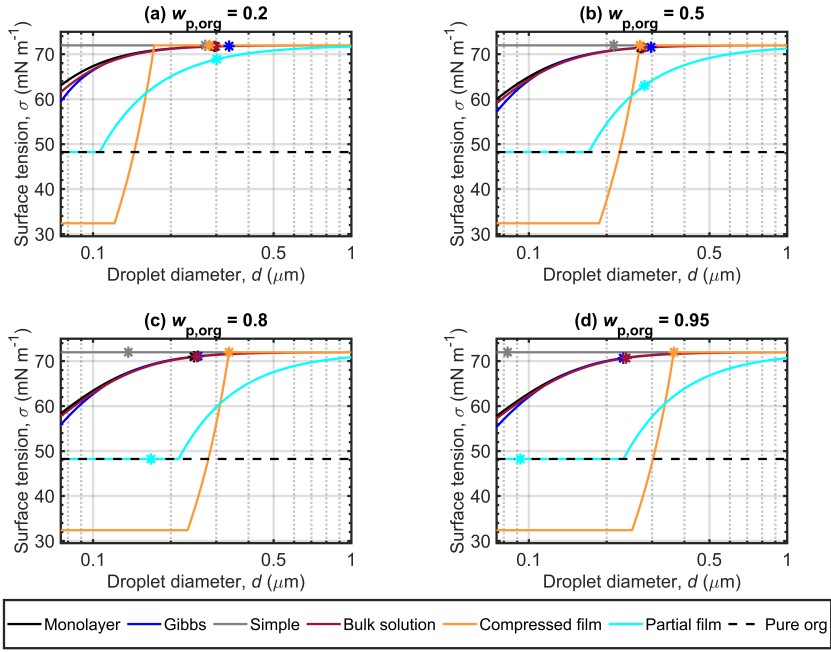

**Figure 3.** Surface tensions of droplets along the Köhler curves, calculated with the different models for dry particles of $D_p = 50$ nm at different malonic acid mass fractions ($w_{\mathrm{p,org}}$). The critical points evaluated for the Köhler curves in Fig. 2 are also marked, and the surface tension of supercooled pure malonic acid estimated though the fit of Hyvärinen et al. (2006) is indicated as a physical lower limit for the droplet surface tension.

in $n^S_{\mathrm{org}}/n^T_{\mathrm{org}}$ to a minimum value before increasing towards unity, as expected. The minimum value observed in Fig. 4(a) is due to the unconstrained organic surface partitioning factor, which does not affect the Köhler curve (Fig. 2) until the droplet surface tension (Fig. 3) begins to increase from its minimum value. The surface tension is being constrained by the model parameter $\sigma_{\mathrm{min}}$, and the fraction of organic molecules partitioned to the droplet surface ($f_{\mathrm{surf}}$) has no effect on the EoS while the constraint applies (see Eqs. (3) - (6)). The Figs. presented by Ruehl et al. (2016) in their supplement for the organic surface

partitioning factor of various organic acids did not show such a minimum, aside from a minor one with particles containing pimelic acid. A likely explanation for this is the different ranges indicated on figure axes in this work (for comparison with other model predictions) and the work of Ruehl et al. (2016).

For Figs. 4(b), 4(c) and 4(d), the surface partitioning factor predicted with Gibbs model is slightly higher than with the monolayer for all but the smallest droplets. The differences between the different malonic acid mass fractions is more noticeable

for predictions with the monolayer model, as the amount of molecules in the surface is constrained by the volume of the molecular monolayer (Malila and Prisle, 2018) while the Gibbs model has no constraint on the extent of surface partitioning. The partitioning factors predicted with Gibbs model are very similar across the droplet size range of the Köhler curves for malonic acid mass fractions $w_{\mathrm{p,org}} = 0.5$, 0.8 and 0.95 and the same can be seen for the compressed film model. At droplet

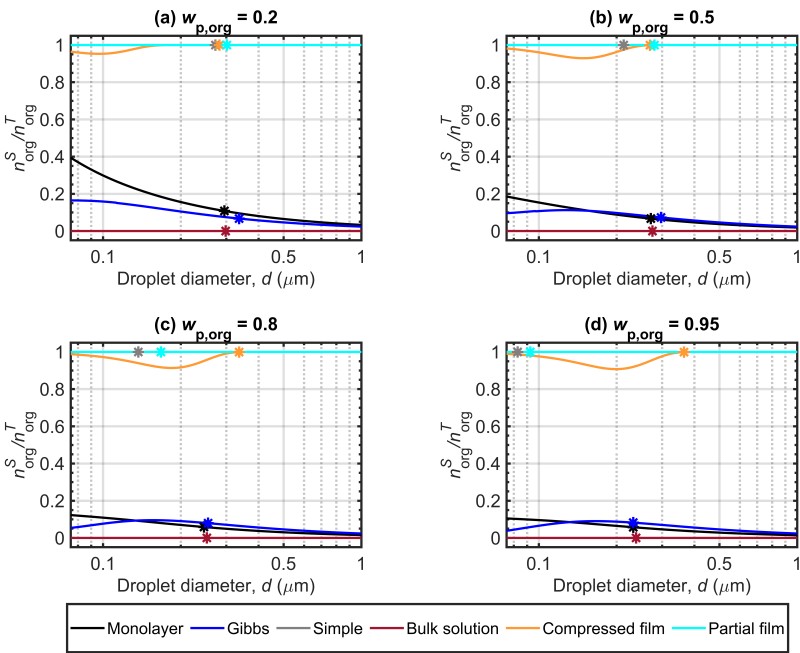

**Figure 4.** Malonic acid surface partitioning factors ($n_{\mathrm{org}}^S / n_{\mathrm{org}}^T$) predicted with the different models along the Köhler curves for dry particles with $D_p = 50$ nm at different organic mass fractions ($w_{\mathrm{p,org}}$). The critical points are also marked.

activation, the partitioning factors predicted here with the Gibbs and monolayer models are similar to the values reported by
Lin et al. (2018) for mixed succinic acid and NaCl particles with the same dry size and $w_{\mathrm{p,org}} = 0.5$. The predicted partitioning
factors with the compressed film model are higher, but comparable to the results of Ruehl et al. (2016), where the values for
malonic acid dry particles of 150 nm and $w_{\mathrm{p,org}} = 0.96$ are always above a minimum of 0.65 (estimated by visual inspection
of Fig. S4 from the supplement of Ruehl et al. (2016)).

## 4 Conclusions

We have compared Köhler model predictions for particles comprising moderately strong organic surfactants, using six different
approaches to describe bulk–surface partitioning in the growing droplets. Specifically, we used the monolayer (Malila and
Prisle, 2018), Gibbs (Prisle et al., 2010), simple (Prisle et al., 2011), compressed film (Ruehl et al., 2016), partial organic
film (Ovadnevaite et al., 2017) and a bulk solution models to predict possible bulk–surface partitioning effects during Köhler
calculations for particles of $D_p = 50$ nm consisting of atmospherically relevant dicarboxylic acids mixed with ammonium
sulphate over a range of compositions. From the Köhler calculations, we evaluated the droplet mixing state in terms of bulk
and surface compositions, droplet surface tension and resulting equilibrium water saturation ratio, as well as the critical point
of droplet activation from the Köhler growth curve maximum and corresponding diameter.

**Table 3.** The critical droplet diameters ($d_c$), supersaturations ($SS_c$) and surface tensions ($\sigma_c$) predicted with the different models in simulations for mixed malonic acid–ammonium sulfate particles of $D_p = 50$ nm at 298.15 K.

| Parameter | $d_c$ (nm) | $SS_c$ (%) | $\sigma_c$ (mN m$^{-1}$) | $d_c$ (nm) | $SS_c$ (%) | $\sigma_c$ (mN m$^{-1}$) | $d_c$ (nm) | $SS_c$ (%) | $\sigma_c$ (mN m$^{-1}$) |
|---|---|---|---|---|---|---|---|---|---|
| $w_{\mathrm{p,org}}$ | Monolayer | | | Gibbs | | | Simple | | |
| 0.2 | 294.97 | 0.46 | 71.73 | 336.43 | 0.42 | 71.81 | 272.19 | 0.51 | 71.97 |
| 0.5 | 271.19 | 0.51 | 71.43 | 297.13 | 0.47 | 71.56 | 213.09 | 0.66 | 71.97 |
| 0.8 | 245.72 | 0.57 | 70.97 | 254.86 | 0.55 | 71.09 | 136.96 | 1.04 | 71.97 |
| 0.95 | 231.77 | 0.6 | 70.65 | 231.77 | 0.61 | 70.68 | 82.72 | 1.88 | 71.97 |
| $w_{\mathrm{p,org}}$ | Bulk solution | | | Compressed film | | | Partial organic film | | |
| 0.2 | 298.22 | 0.46 | 71.73 | 282.31 | 0.48 | 71.97 | 301.5 | 0.45 | 68.87 |
| 0.5 | 275.19 | 0.5 | 71.43 | 269.22 | 0.59 | 71.97 | 280.26 | 0.49 | 63.04 |
| 0.8 | 252.08 | 0.56 | 71.0 | 336.43 | 0.58 | 71.97 | 167.43 | 0.54 | 48.24 |
| 0.95 | 237.77 | 0.59 | 70.68 | 364.58 | 0.57 | 71.97 | 92.64 | 1.05 | 48.24 |

When the mass fraction of surface active organic is small ($< 50$ %), the predicted Köhler growth curves and critical supersaturation values for droplet activation are similar between the models. For particles with high mass fractions ($> 80$ %) of organic, significant differences begin to appear. The simple and partial film models start to predict increased $SS_c$ and decreased $d_c$ compared to the rest of the models. The full partitioning models (monolayer, Gibbs, and compressed film) all predict similar critical supersaturations as the bulk solution model for the investigated dicarboxylic acid systems, although the compressed film model predicts larger critical droplets. Despite these overall similarities, there are however large differences between the different models in the predicted degree of surface partitioning of the organic component. For the simple, partial film and bulk solution models, the degree of partitioning is included in the model assumptions. Between the full partitioning models, the degree of organic surface partitioning predicted with the compressed film model is significantly higher than with either the Gibbs or surface monolayer models. It was not a priori expected that the compressed film and the surface monolayer models would predict similar values of $SS_c$ for a given aerosol system. For the compressed film model, all surfactant is predicted to have partitioned to the droplet surface at the point of activation. The bulk solution model does not consider bulk–surface partitioning and all surfactant solute remains in the bulk phase which constitutes the full volume of the droplet. This seems contradictory and may reflect a lack of model robustness across droplet conditions outside the limited range of systems and conditions for which the models have been directly validated by measurement. Different predictions of $SS_c$ for the same particle systems, or predictions of similar $SS_c$ with nearly opposite degrees of organic bulk–surface partitioning, leads to uncertainty regarding how well the underlying phenomena are represented with each of the models in question.

For all mass fractions of the surface active organics in the particles, the different models predict a range of different surface tension curves for the growing droplets, as expected with the variety of applied equations and assumptions governing surface tension evolution in the droplets. The surface monolayer and Gibbs partitioning models use the same surface tension

parametrizations as the bulk solution model and all predict surface tension curves of similar shape, reflecting similar mixing states of the growing droplets. For the simple partitioning model, droplets are predicted to have a constant surface tension, while the compressed and partial organic film models each predict distinct surface tension curves, reflecting the underlying assumptions regarding both the bulk–surface partitioning and surface tension equation of state. The predicted droplet surface tensions at the point of activation are comparable for all models at small organic fractions in the particles, but differences between the models increase with the organic fraction. The partial organic film model consistently predicts the lowest surface tension at droplet activation, for some particle compositions as low as the surface tension for the pure organic. The largest droplet surface tension depressions observed with the partial film model in the present work are larger than was observed by Ovadnevaite et al. (2017). Furthermore, droplet activation at high organic mass fractions is here predicted to occur before the surface film breaks. The highest organic mass fraction in the particles is also larger in the present study than in the work of Ovadnevaite et al. (2017), which directly contributes to the predicted surface tension depression. The other models predict only moderate or no surface tension depression in droplets at activation. Our predictions using the compressed film model agree well with previous studies (e.g. Ruehl et al., 2016; Forestieri et al., 2018) in this regard, as do the Gibbs and monolayer model predictions to previous results of these models for particles containing surface active compounds of comparable strength to those used in the current study (e.g. Malila and Prisle, 2018; Lin et al., 2018; Prisle et al., 2019).

Among the models used in this work, the Gibbs, monolayer, and compressed film models evaluate the progression of the bulk–surface partitioning equilibrium with mixing state as the droplets grow, whereas the simple partitioning and partial organic film models rely on the simplifying assumption that the organic is completely partitioned to the droplet surface. We see that for particles where the fraction of organic is not too large, the latter models can still yield similar results as the comprehensive models, but underlying assumptions may become increasingly misrepresentative as the fraction of surface active organic in the particles becomes larger. Regarding the comprehensive partitioning models, the Gibbs and monolayer models predict similar droplet properties at activation as the bulk solution model, due to the modest degree of organic surface partitioning at activation, which seems realistic for dilute solutions of a surface active compound of moderate strength, and because models use the same surface tension parametrizations based on independent measurements. The compressed film model on the other hand uses a surface tension equation of state with parameters which are obtained by fitting to experimental droplet growth curves similar to those predicted and predicts very strong surface partitioning and surface tension depression in the growing droplets.

Average supersaturations in low-level clouds range from 0.1 % to 0.4 % (e.g. Politovich and Cooper, 1988), however higher values between 0.7 % and 1.3 % can be reached during strong convection (e.g. Yang et al., 2019; Siebert and Shaw, 2017). The predicted Köhler curves presented here for 50 nm particles comprising each of the organic acids and ammonium sulphate could provide insights to the cloud activation potential of secondary organic aerosols in the Aitken mode. With the six modeling approaches, we here predict critical supersaturations from 0.42 % to 1.88 % for increasing amounts of malonic acid in the particles, between 0.42 % and 1.9 % for succinic acid particles (see section S1 of the supplement), and between 0.43 % and 1.96 % for glutaric acid particles (see section S1 of the supplement). This indicates that activation of droplets similar to those studied here will mostly occur during strong convection. Since the Aitken mode dominates number concentrations in atmospheric secondary organic aerosols (e.g. Vaattovaara et al., 2006; Kulmala et al., 2016), fluctuations in predicted droplet

numbers will translate into variations in cloud microphysics and cloud optical properties when the aerosol loading is high. Other large-scale effects could be related to the ice nucleation ability of organic aerosols in high-level clouds. With surface solutions enriched in organic compounds, they could play a significant role in the heterogeneous ice formation in cirrus cloud (Wise et al., 2010; Piedehierro et al., 2021).

Overall, this comparison of different model predictions for surface active aerosols clearly highlights the need for further experimental validation of the different bulk–surface partitioning models across a wide range of particle mixtures and conditions, before any of the models is used as basis for broad generalizations of results to atmospheric processes. Our results highlight how the different models may predict similar activation properties but very different degrees of organic bulk–surface partitioning in droplets. Comparison to experimental $SS_c$ values is valuable for confirming critical droplet properties predicted with the different models for a range of conditions, and for surfactants of varying strengths, but will not offer insight into the exact role and dynamics of bulk–surface partitioning in the droplet activation process. Recently, Bzdek et al. (2020) presented the first direct experimental observation of the impact of size-dependent bulk–surface partitioning in droplets suspended in air. Such studies however remain highly elaborate and sparse, and therefore creating robust models and comparing different models across a variety of different systems can offer insight into the process and guide future experimental studies. In this work, we predict very similar $SS_c$ values using the Gibbs, monolayer, and compressed film partitioning models and the bulk solution model, which are furthermore similar to the experimental $SS_c$ values, despite significant differences in predicted organic bulk–surface partitioning between the models. Given that the bulk solution model does not represent demonstrated effects of organic bulk–surface partitioning in small droplets and the compressed film model can predict unphysical droplet surface tensions, this suggests that the Gibbs and monolayer partitioning models are currently the preferable options for modeling droplet growth and activation of aerosol systems comprising moderately strong surface active compounds. A similar comparison between the different droplet models for more strongly surface active particle components is the topic of future work. For such aerosol systems, conclusions regarding the different models may differ significantly from those of the present work.

*Data availability.* Output data of the different models is available at https://doi.org/10.5281/zenodo.5832468 (Vepsäläinen et al., 2022).

*Author contributions.* SV adopted the models for the study, did the model simulations, and performed the analysis of model results with assistance from SMC, JM and NLP. SV wrote the original manuscript draft and made the visualizations with NLP and assistance from SMC and JM. SV and NLP wrote the revised manuscript and authors' response to reviewers with input from co-authors. NLP conceived the project and methodology, was responsible for the supervision and project management and secured the funding for the work.

*Competing interests.* There are no conflicts to declare.

*Acknowledgements.* We thank Chris Ruehl and Chris Cappa for providing and discussing their codes of the compressed film model. We also thank Andreas Zuend for providing the surface coverage and surface tension calculation routine used for the partial organic film model. This project has received funding from the European Research Council (ERC) under the European Union's Horizon 2020 research and innovation programme, Project SURFACE (Grant Agreement No. 717022). The authors also gratefully acknowledge the financial contribution from the Academy of Finland (Grant Nos. 308238, 314175, and 335649).

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
