# Peer review of "Comparison of six approaches to predicting droplet activation of surface active aerosol. Part 1: moderately surface active organics"

_Atmospheric Chemistry and Physics, 2021_

## Author Comment (AC1)

**Author response to reviewers' comments**

Sampo Vepsäläinen[1], Silvia M. Calderón[1,2], Jussi Malila[1], and Nønne L. Prisle[1,3]

[1]Nano and Molecular Systems Research Unit, University of Oulu, P.O. Box 3000, FI-90014, Oulu, Finland
[2]Finnish Meteorological Institute, P.O. Box 1627, FI-70211, Kuopio, Finland
[3]Center for Atmospheric Research, University of Oulu, P.O. Box 4500, FI-90014, Oulu, Finland

**Correspondence:** Nønne L.Prisle (nonne.prisle@oulu.fi)

We thank both reviewers for their careful revision of our manuscript and constructive comments. Below we provide our responses in a point-by-point manner. The reviewers' comments are reproduced in *italics*, our responses in blue and quotes from the revised manuscript in **red bold** font. In addition, we have made a few minor edits to the manuscript in places not directly indicated by the reviewers, to further clarify the points.

**1 Referee 1**

**1.1 General comments**

*The paper focuses on cloud droplet activation models with different approaches to account for surfactant effects, which is relevant topic for ACP. The models are presented and used accurately. The language is fluent and precise, and the overall presentation quality is good. However, the paper is mostly based on previously developed approaches that has been tested elsewhere. Partly for this reason the main aim and the conclusions are also somewhat unclear, so they need to be emphasized (details below) before the manuscript can be accepted.*

**1.2 Specific comments**

*Introduction: the extensive review of previous research shows that this is not a new topic, so it would be important to point out the open research questions which will be addressed in this study. Specifically, what is the added value of comparing multiple models here when they have been examined elsewhere sometimes with greater details and often with larger amount experimental data? Is it important that the models predict different partitioning equilibrium when the predicted critical supersaturations are similar? Regarding the previous studies mentioned in the introduction, are there clear issues with the models that can be fixed here?*

The phenomena related to surface activity are not new and different models have been presented in the literature to account for them. Comparisons against $SS_c$ data have also been done for a limited set of systems (e.g. Prisle et al., 2008, 2010; Ruehl et al., 2016; Lin et al., 2018; Davies et al., 2019). Although a few of the models have been compared against each other (e.g. Lin et al., 2018), a full comparison of the predictions with the different most commonly used models has so far not been made. To our knowledge, these models have also not been compared to experimental data for the same system and conditions.

This comparison of the available models for the same systems and conditions and evaluation of their mutual robustness for predicting the cloud droplet activation and specific role of surface activity is the main novelty of this work. The degree to which model predictions agree for a common system and conditions indicates the confidence in the ability of the models to describe the droplet activation and capture the role of aerosol surface activity in the process. Any differences between the model predictions could indicate systems and conditions where some or all of the models are not internally robust despite previous validation against a few experimental systems. When the different models predict different droplet activation properties, for example when models are applied for systems and conditions for which they have not been explicitly validated and where experimental data is currently not available, it is not clear which model represents the actual system more closely. This has also been clarified in the Conclusions section.

The importance of the different extents of organic bulk–surface partitioning predicted with the different models depends on the purpose. It is less important for comparing the model predictions to available experimental critical supersaturation or for obtaining values of $SS_c$ to be used elsewhere. However for understanding bulk–surface partitioning and the dynamics involved between partitioning and dilution of growing and activating droplets, it is quite important. This work in particular assesses differences between predictions of available bulk–surface partitioning models, in this case for several common organic aerosol compounds of moderate surfactant strength. It is however difficult to say whatever the same models would predict similar $SS_c$ values consistently with different simulation conditions and for other systems without further investigation. In addition, the detailed understanding of surfactant effects could be important for other atmospheric processes (Prisle, 2021), for example aqueous surface chemistry, in addition to cloud droplet activation.

This manuscript presents a comparison of current state-of-the-art models available in the literature and in use of the community. It is outside the scope of the present study to directly evaluate the validity of the underlying assumptions of the models. For exhaustive documentation of these models, we refer to the original publications. Furthermore, our aim is to compare the models as they are currently presented and used and we do not attempt to significantly amend or improve the models in the present work.

The following text has been added to the introduction to expand and explain the motivation and value of the current work:

**"The models each offer a different description of the phenomena related to surface partitioning and surface tension in small droplets. Each of the models have previously been compared with experimental data (e.g. Prisle et al., 2008, 2010; Ruehl et al., 2016; Lin et al., 2018; Davies et al., 2019) for a limited set of droplet systems and conditions. However, critical supersaturation ($SS_c$) data can only validate model predictions of the critical point of droplet activation and does not allow for a direct assessment of predictions of the bulk–surface partitioning in the droplets. The flexibility and robustness of the models, in terms of their ability to describe systems and conditions other than those used for their validation so far remains an open question. To our knowledge, the extent to which all the available models predict consistent droplet growth and activation properties for the same surface active aerosol systems and conditions has so far not been investigated. When models are applied in conditions for which they have not been directly validated, as for example across a broad range of aerosols and conditions in large scale simulations (Prisle et al., 2012; Lowe et al.,**

**2019), it is important to know whether significant differences in predicted droplet activation and cloud properties could occur, depending on the choice of surface activity model."**

60     *A separate companion study is mentioned in the introduction (line 116), but could these be merged? It looks like introduction and theory & methods sections already contain relevant information for the strong surfactants. Combining the results could show which models are suitable for both strong and moderately strong surfactants. In addition, this would expand the range of modelled systems currently focusing on malonic acid-ammonium sulfate.*

    The Introduction, Theory and Methods sections already cover a significant amount of information for strong surfactants

65 because historically these compounds have been the most studied in the context of aerosol surface activity and cloud droplet activation. Previous studies indicate that the role of surface activity in cloud droplet activation may be very different for strongly and moderately surface active aerosols (see e.g. summary by Prisle (2021)). The present manuscript (and supplement) is already of significant length and presents results for aerosol systems comprising three different carboxylic acids mixed with ammonium sulphate at a range of different mixing ratios. To accommodate simulations for particles comprising also the strong surfactant,

70 new sections would need to include the consideration of CMC and additional compound properties including the associated parameterizations. The results for strongly surface active aerosols would need a range of additional tables and figures, as well as analysis and discussion of related findings and the comparison between strong and moderately strong surfactant properties. The conclusions regarding the model comparison may differ significantly between the strong and moderately strong surfactants and highlight different aspects of the models and effects of surface activity on cloud droplet activation. Therefore, we prefer to

75 treat model comparison for these different types of aerosol systems in separate articles. However, as the models used for the companion article are the same as presented in this work, used for different systems and conditions and with the treatment of CMC for the strong surfactants, the title of the current manuscript has been changed to include "Part 1" while the manuscript concerning strong surfactants will be titled "Part 2".

    In addition, as a response to the comment from referee 2 about lines 71-74 of the original manuscript, some of the text

80 relating to strong surfactants has been shortened.

    *Line 106-113: "We quantify surfactant … dicarboxylic acids (e.g. Booth et al., 2009; Hyvärinen et al., 2006; Ruehl et al., 2016)." seem to be misplaced; Sect. 2 could be more suitable for the details.*

    The details have been moved to Section 2.

    *Line 162: What do you mean by the "method of solving the Gibbs adsorption equation"? Numerical method (this should not*

85 *have noticeable effects on results)?*

    This was a poor choice of words. The sentence was intended to reflect on the different specific boundary conditions that are used to solve the equation. For example, the boundary condition that the molar ratio of water and salt is the same in the bulk and surface phases.

    The text has been rephrased as:

90 **"the specific boundary conditions applied when solving the Gibbs adsorption equation"**

    *Lines 181 and 231: The layer is insoluble, but it doesn't stop water vapor, right? This should be clarified because there are studies where such layer is impermeable.*

Clarification has been added:

Line 181: **"It approximates organic partitioning ... an insoluble layer that is assumed to not affect the kinetics of water condensation–evaporation equilibrium."**

Line 231: **"The film is assumed to not affect the kinetics of water condensation–evaporation equilibrium."**

*Line 187: "Predictions with the comprehensive…" indicates that surface tension depends on surface concentration, but should it depend on bulk concentration?*

All the used models assume thermodynamic equilibrium and the droplet surface tension can be described in terms of either the bulk or surface concentration of surfactant, which are in equilibrium between the phases. Indeed, the surface tension depends on the bulk concentration for the Gibbs model due to the possible partial bulk–surface partitioning of the organic i.e. surfactant can remain in the bulk affecting the bulk composition. The simple model assumes all surfactant is partitioned to the droplet surface (none remains in the bulk), and the surface tension is equal to that of pure water for all calculations.

*Line 241: Why is the minimum surface thickness dependent on droplet size in your simulations (Ovadnevaite et al. (2017) used two different constant values)?*

The monolayer and partial organic film models differ in their representation of the minimum surface thickness. In the original model of Ovadnevaite et al. (2017), two separate constant values were assumed for the droplet surface thickness. In this work, the minimum surface thickness used for calculations with the partial organic film model is the same as the corresponding thickness calculated with the molecular monolayer model of Malila and Prisle (2018). The monolayer model explicitly predicts a variable surface thickness that depends on the composition of the droplet surface solution, and therefore depends on droplet geometry. Limiting values for surface thickness are given by the equivalent droplet diameter of a sphere holding a volume equal to the molecular volume of pure droplet constituents. The original partial organic film model of Ovadnevaite et al. (2017) is designed to assume a single constant value of the surface thickness, which is a simplification and likely not physically representative for growing and diluting droplets and for droplets comprising a range of different surface active compounds. Since the monolayer model explicitly calculates surface thickness, we here use these values to enhance consistency between the results of the different models.

*Theory & Methods: Why only some models use parametrized solution density and water activity (Table 1)? Is this causing differences seen in Fig. 1?*

The calculations of the amount of water in the droplets are following the model structures given in the respective original references for each model. We have included an explanation of the water activities in section 2.1 of the revised manuscript in response to a comment from referee 2 about the activities and therefore we only address the calculation of the amount of water in the droplets here. The full response regarding water activities used in our calculations can be found in the Section 2.1 of the response.

The $a_w$ and Ke terms are shown for malonic acid–ammonium sulphate particles with $D_p = 50$ nm in Figs. 1 and 2 below (in this response).

Differences between the predicted Köhler curves shown in Fig. 1 of the original manuscript are of course in part due to differences in the underlying structure and assumptions of the different models. Illustrating these effects across the Köhler

[Figure]

**Figure 1.** The water activity calculated with the different models for dry malonic acid–ammonium sulphate particles with $D_p = 50$ nm. Each panel shows curves for particles with a different malonic acid mass fraction ($w_{p,org}$). The critical points are marked on each curve.

curve and for a range of particle compositions is the point of the current work. Figs. 1 and 2 below show that there are large differences in both the Raoult and Kelvin terms of the Köhler curves for malonic acid–ammonium sulphate particles at $D_p = 50$ nm, as expected with significantly different assumptions of the different models regarding the bulk–surface partitioning, the droplet compositions and surface tensions. The predictions of $a_w$ can clearly be divided into two groups in Fig. 1(c-d), reflecting the different evaluations of compositions of the droplet bulk. The simple, partial organic film and compressed film models are grouped together because the first two assume all the organic to partition to the surface and the compressed film model predicts a very strong surface partitioning at given conditions. The Gibbs and monolayer models predict a significant amount of surfactant to remain in the droplet bulk, and the bulk solution model does not evaluate bulk–surface partitioning at all. The Kelvin terms in Fig. 2 behave consistently between the models across the different mass fractions of organic in the particles, which is attributed to the similar predicted droplet surface tensions.

The following text has been added to section 2.1 of the revised manuscript to address the calculation of the amount of water in the droplets:

**" The calculation of the amount of water in each droplet uses composition dependent density of the droplet solution (indicated in Table 1 as *iterative* calculations) when required by the model (bulk solution and monolayer models). For all other models (Gibbs, simple, compressed film and partial organic film models), additive volumes of water and the dry particle components were used as indicated in the original model descriptions. "**

[Figure]

**Figure 2.** The Kelvin terms calculated with the different models for dry malonic acid–ammonium sulphate particles with $D_p = 50$ nm. Each panel shows curves for particles with a different malonic acid mass fraction ($w_{\mathrm{p,org}}$). The critical points are marked on each curve.

*Line 315: "This is most likely an effect. . . " seems vague. You could confirm this by swapping parameterizations. There are a few other cases where "likely" (lines 321, 354, 373, etc.) could be confirmed.*

The word "likely" is sometimes used a bit too generously by some of us non native English writers. We have revised the sections mentioned, and added detail to certain cases as shown below. The sentence on line 315 of the original manuscript was also confirmed by running the two models with different parameterizations. From line 354, the text has been revised as:

**"This is an effect of the closely similar conditions predicted with the two models after the critical point of the Köhler curve. The compressed film model predicts all organic to have partitioned to the surface by the point of activation and therefore the surface tension is equal to that of pure water for droplets after this point. The only differences between models for droplets in this size range are due to the Raoult terms, which are also very similar (with differences on the order of $10^{-5}$)."**

From line 373, the sentence has been investigated and uncertainty has been removed. We revise as:

**"These differences reflect the surface activity of the organic component with pollenkitt being a more strongly surface active substance than malonic acid."**

*Line 322-323: For me it doesn't look like the partial organic film model predicts a "considerably" lower SSc (it is about the same as for the four other models) and neither is the dc "considerably" smaller (it is the second smallest).*

Thank you for noticing. The sentence has been corrected to:

**"The partial organic film model predicts $SS_c$ comparable to the other models, except for the simple model. After the simple model, values of $d_c$ predicted with the partial film model are the smallest among the different models."**

*Line 327: "falls between the monolayer and simple model predictions" could be true, but it looks like all other SSc values except that from the simple partitioning model are similar.*

The sentence has been changed to:

**"The Köhler curve calculated with the compressed film model has a distinct shape, compared to the other models, but the $SS_c$ is similar to the other model predictions excluding the simple model. The compressed film model predicts the largest $d_c$ value in Fig. 2(c)."**

*Lines 334-336: "The sensitivity analysis shows . . . predictions are quite robust . . . respect to relatively large variation ..." is a vague statement.*

The text has been changed to include more detail about the variations that affect the critical point:

**"The sensitivity analysis shows that predictions of the critical point with the compressed film model are stable with respect to relatively large variations in the model parameters for particles containing malonic and glutaric acid. The critical point destabilises (moves to the second local maxima) at perturbations of 9-52 % depending on the model parameter in question. The predictions for succinic acid particles are more sensitive to variations in model input parameters, as the perturbations required to move the critical point are only between 3-6 % for the different model parameters."**

*Lines 344-347: This sentence is one example which shows that these models have been examined in previous studies, so emphasizing new findings is especially important. The same comment applies to sections describing surface tension and organic bulk-surface partitioning.*

The main novelty of the work is the comparison of all the different models for the same aerosol systems under the same conditions. This has also been clarified in the introduction of the revised manuscript. We have made the following changes to the text to clarify the point of the reviewer.

Line 344: "The high SSc values are an effect of the assumed complete partitioning of the organic to the surface and is in line with observations by Prisle et al. (2011) for how the simple model predictions differ from both experiments and detailed model predictions (Prisle et al., 2010) at high organic mass fractions, even with fatty acid salts and SDS, all relatively strong surfactants."

Lines 344-347 have been reformulated. The explanation for the high $SS_c$ values has been combined into another correction relating to lines just above 344 of the original manuscript. Emphasis on the findings of the present work has been added:

**"These findings for moderately surface active organics are in line with observations by Prisle et al. (2011) for strong surfactants such as fatty acid salts and SDS, that predictions with the simple model differ from both experiments and the detailed Gibbs model (Prisle et al., 2010) at high organic mass fractions."**

In the surface tension section, a sentence emphasizing observations new to this work has been added to the revised manuscript:

**"However, Ovadnevaite et al. (2017) did not predict activation to occur before breaking of the organic film for roughly $w_{\mathrm{p,org}} = 0.5$ and at $D_p = 175$ nm or at $D_p = 41$ nm, as seen here in Figs. 3(c) and (d)."**

The following observation has added to the organic bulk-surface partitioning section of the results:

**"The Figs. presented by Ruehl et al. (2016) in their supplement for the organic surface partitioning factor of various organic acids did not show such a minimum, aside from a minor one with particles containing pimelic acid. A likely explanation for this is the different ranges indicated on figure axes in this work (for comparison with other model predictions) and the work of Ruehl et al. (2016)."**

*Section 3.1.2: Is the shape of droplet growth curve important for other than SSc? This is not clear based on the introduction. Also, is the first the first part of the Köhler curve (before the film breaks) from the compressed and partial film models physically meaningful, or is it based on assumptions, parameters or extrapolation? Regarding the model parameters, the supplement shows sensitivity analysis for the compressed film model, but this is not done for the partial film model (or the other models); why and what would the results show?*

The shape of the droplet growth curve is key to understand the response of a growing particle to changes in the relative humidity, including not only droplet activation but also water uptake if the particle remains as interstitial aerosol. For example, a sharper slope of the water saturation ratio with respect to droplet size reflects stronger sensitivity of droplet growth to variations in atmospheric conditions (i.e. activation vs. deactivation during updrafts/downdrafts inside an air parcel). If a Köhler model produces sharper growth curve activation peaks, it may introduce additional variability to simulations of cloud formation that may obscure the underlying phenomena. The shape of the Köhler curve can also reveal information about specific events during the growth of the droplet, such as the breaking of the surface film with the partial film model. For droplets comprising strong surfactants, reaching the CMC can also be seen in the shape of the Köhler curve (see e.g. Prisle (2021)).

For the partial film model, the first part of the Köhler curve corresponds to droplets where the surface film is intact, and the surface tension is that of the pure organic component. Therefore it does have physical meaning. The following has been added in section 3.1.1 of the revised manuscript discussing the instance when the organic mass fraction in the particles is $w_{\mathrm{p,org}}$ = 0.8:

**"The critical point of activation predicted with the partial film model occurs at a droplet size where the surface film is intact. This was not observed in previous studies by Ovadnevaite et al. (2017) for their particles comprising a surrogate organic mixture and ammonium sulphate with $w_{\mathrm{p,org}} = 0.5$ and $D_p = 175$ nm or $D_p = 41$ nm, and neither by Davies et al. (2019) for particles containing suberic acid and ammonium sulphate with $w_{\mathrm{p,org}} = 0.49$ at $D_p = 100$ nm or $D_p = 40$ nm."**

The explanation for compressed film model is more complex. Since the next three referee comments deal with the same subject, our response is distributed between these comments.

The sensitivity analysis was initially performed only for the compressed film model, because this model predicts a prominent second local maximum for $w_{\mathrm{p,org}} = 0.95$ which is not seen for the other models to the same extent. The other most prominent case is with the partial film model for $w_{\mathrm{p,org}} = 0.8$ where the two maxima occur before and after the surface film breaks, respectively. However, the compressed film model is different from the other models in that the unique model parameters are acquired through fitting to Köhler curve data, and are available in literature only for a few systems (e.g. Ruehl et al., 2016; Forestieri et al., 2018). Furthermore, in line with previous studies these model parameters are treated like physical constants

that are assumed to be valid across the full range of conditions realized by growing and activating droplets. Without this assumption, it would not be possible to perform calculations with the compressed film model for the organic compounds relevant to the current work, because the parameter variations are not determined and the simulation conditions ($D_p$, $w_{p,org}$ etc.) are different from those of the original parameter fitting in Ruehl et al. (2016). This assumption has also been made in other applications of the model. The sensitivity analysis presented in this work highlights the implications of this assumption with respect to conditions where large changes to droplet critical properties seem most likely, as seen e.g. in Fig. 1(d) of the original manuscript.

A comprehensive sensitivity analysis including all the models is now provided in the supplement.

*Section 3.2: The same comment as above. Surface tension from the compressed film model is lower than the "lowest physically realistic value for the droplet surface tension" (line 398), what does this mean? Related to the first part of the Köhler curves, it is written in the manuscript that the compressed film model uses extrapolation (line 413: "...extrapolation of the model parameters outside of their validity region..."), so should this part of the curve considered when determining the SSc?*

The following text has been added to section 3.2 to address the lowest surface tensions predicted with the compressed film model:

**"In addition, the surface tension curves predicted with the compressed film model start at a minimum surface tension value below that of the measured surface tension of supercooled malonic acid. This is a result of the compressed film EoS presented by the original work of Ruehl et al. (2016) and given here in Eq. (6). The lower limit for the surface tension given by the equation is constrained by the model parameter $\sigma_{min}$. The parameter is a fitted model parameter, which does not a priori have a physical interpretation and doing so may lead to unrealistic values of physical parameters as seen for the droplet systems investigated here."**

We consider this to be a limitation of the compressed film model framework which should be taken duly into account in its applications. The compressed film model parameters used in the present work were specifically fitted by Ruehl et al. (2016) to create a Köhler curve matching experimental observations of droplet growth, mainly before the critical point of activation, and therefore the whole curve should be considered. However the fitted parameters are assumed to be physical constants that can be applied for a range of dry particle and droplet sizes and compositions. In line 413 when discussing Fig. 1(a) of the original manuscript we ascribe this assumption as a potential reason for the disparity in the predicted behaviour of the critical point compared to Ruehl et al. (2016) i.e. "...extrapolation of the model parameters outside of their validity region...". We rephrase to:

**"... since the model parameters are fitted to experimental observations for particles with high organic mass fraction and growing droplets in the range before the critical point of activation and are therefore not constrained by these measurements across all droplet states realized in our calculations."**

However, using the compressed film model Forestieri et al. (2018) also observed a deviation from previously predicted behaviour where the critical point for particles containing oleic acid did not match the droplet surface tension increasing to the value of pure water. The following sentence has been added to the relevant section in the revised manuscript as further clarification:

**"Forestieri et al. (2018) ascribed this behaviour to originate from the larger molecular volume of oleic acid, compared to the dicarboxylic acids investigated by Ruehl et al. (2016)."**

This however could not be the reason for the critical size of the droplet not matching the size of the droplet when droplet surface tension increases to the value of pure water in the present work (as the organic particle component does not change between the panels of Fig. 1 of the original manuscript). The subject continues with the next response.

*Section 3.3: The same comment as for Sects 3.1.2 and 3.2. The authors write that the compressed film model predict surface tensions that "has no physical meaning for the Köhler curve (Fig. 1) until the surface tension (Fig. 2) starts increasing" (lines 457-458), so what happens if SSc is seen before this point (Fig. S8(d))?*

Our reasoning for this statement can be seen from Eq. (3) - (6) of the original manuscript, where the fraction of organic molecules partitioned to the surface $f_{\text{surf}} = n_{\text{org}}^S / n_{\text{org}}^T$ does not affect the EoS presented in Eq. (6), before the surface tension starts increasing. Additional clarification has been added to section 3.3, and the expression specified to better explain what was intended:

**"The minimum value of in Fig. 4(a) is due to the unconstrained organic surface partitioning factor, which does not affect the Köhler curve (Fig. 2) until the droplet surface tension (Fig. 3) begins to increase from its minimum value. The surface tension is being constrained by the model parameter $\sigma_{\text{min}}$, and the fraction of organic molecules partitioned to the droplet surface ($f_{\text{surf}}$) has no effect on the EoS while the constraint applies (see Eqs. (3) - (6))."**

Figure S8(d) is part of the sensitivity analysis provided for the compressed film model, where the model fitting parameters have been varied to investigate the possibility of the critical point moving to the second prominent local maximum. Therefore this occurrence of $SS_c$ being before the surface tension starts increasing happens under extreme conditions. The following has been added to section S3.1 of the supplement to the revised manuscript (the sensitivity analysis, section S2.4 in the original manuscript):

**"In the predicted Köhler curves for particles containing each of the investigated organic acids (Figs. 2, S1 and S4), there is a prominent second local maximum at $w_{\text{p,org}} = 0.95$ (panel (d) of the Figs. 2, S1 and S4). The sensitivity analysis consists of varying the compressed film model specific parameters $A_0$, $\log_{10} C_0$, $m_\sigma$ and $\sigma_{\text{min}}$ to investigate whether the critical point shifts from the first to the second local maximum, which resides in a droplet region where the model specific minimum droplet surface tension is realized. Outside of this sensitivity analysis, results of each simulation presented in this work, as well as by other studies (e.g. Ruehl et al., 2016; Forestieri et al., 2018), predict that all surfactant has partitioned to the droplet surface at the point of activation, and that the droplet surface tension has reached that of pure water (or a value very close to it). If the critical point is predicted to occur at droplet conditions significantly different from these, it indicates lack of robustness in the fitting of the model parameters."**

*Conclusions: Are the "large differences between the different models in the predicted degree of surface partitioning of the organic component" mostly related to the model assumptions (this question applies to surface tension as well)? It would be useful to point out possible unexpected findings. Previous studies, which are mentioned in the previous sections, have made similar findings, so what is new in this study? In addition to listing the differences between model predicted surface tensions and partitioning factors, you could mention which of these predictions are closer to reality and what do they mean from micro-*

*scopic point of view. You cloud also mention if current findings have any large-scale effects for cloud activation or clouds in*
300 *general.*

We again emphasize that the main novelty of the present work is the comparison of predictions with the different models for representing organic aerosol surface activity for the same particle systems and under the same conditions. The following comment on the unexpected findings regarding the varying levels of predicted bulk–surface partitioning of the organic compound in growing droplets has been added to the Conclusions:

305 **"It was not a priori expected that the compressed film and the bulk solution models would predict similar values of $SS_c$ for a given aerosol system. For the compressed film model, all surfactant is predicted to have partitioned to the droplet surface at the point of activation. The bulk solution model does not consider bulk–surface partitioning and all surfactant solute remains in the bulk phase which constitutes the full volume of the droplet. This seems contradictory and may reflect a lack of model robustness across droplet conditions outside the limited range of systems and condi-**
310 **tions for which the models have been directly validated by measurement. Different predictions of $SS_c$ for the same particle systems, or predictions of similar $SS_c$ with nearly opposite degrees of organic bulk–surface partitioning, leads to uncertainty regarding how well the underlying phenomena are represented with each of the models in question."**

We have also added more detail to the Conclusions about droplet surface tension:

"For all surface active organic mass fractions in the particles, the different models predict very different surface tension curves
315 for the growing droplets."

**"For all mass fractions of the surface active organics in the particles, the different models predict a range of different surface tension curves for the growing droplets, as expected with the variety of applied equations and assumptions governing surface tension evolution in the droplets."**

These references to previous studies have also been added to the same sections:

320 **"The largest droplet surface tension depressions observed with the partial film model in the present work are larger than was observed by Ovadnevaite et al. (2017). Furthermore, droplet activation at high organic mass fractions is here predicted to occur before the surface film breaks. The highest organic mass fraction in the particles is also larger in the present study than in the work of Ovadnevaite et al. (2017), which directly contributes to the predicted surface tension depression."**

325 and

**"Our predictions using the compressed film model agree well with previous studies (e.g. Ruehl et al., 2016; Forestieri et al., 2018) in this regard, as do the Gibbs and monolayer model predictions to previous results of these models for particles containing surface active compounds of comparable strength to those used in the current study (e.g. Malila and Prisle, 2018; Lin et al., 2018; Prisle et al., 2019)."**

330 The following has been added to the Conclusions regarding larger scale cloud effects:

**"Average supersaturations in low-level clouds range from 0.1 % to 0.4 % (e.g. Politovich and Cooper, 1988), however higher values between 0.7 % and 1.3 % can be reached during strong convection (e.g. Yang et al., 2019; Siebert and Shaw, 2017). The predicted Köhler curves presented here for 50 nm particles comprising each of the organic acids**

**and ammonium sulphate could provide insights to the cloud activation potential of secondary organic aerosols in the Aitken mode. With the six modeling approaches, we here predict critical supersaturations from 0.42 % to 1.88 % for increasing amounts of malonic acid in the particles, between 0.42 % and 1.9 % for succinic acid particles (see section S1 of the supplement), and between 0.43 % and 1.96 % for glutaric acid particles (see section S1 of the supplement). This indicates that activation of droplets similar to those studied here will mostly occur during strong convection. Since the Aitken mode dominates number concentrations in atmospheric secondary organic aerosols (e.g. Vaattovaara et al., 2006; Kulmala et al., 2016), fluctuations in predicted droplet numbers will translate into variations in cloud microphysics and cloud optical properties when the aerosol loading is high. Other large-scale effects could be related to the ice nucleation ability of organic aerosols in high-level clouds. With surface solutions enriched in organic compounds, they could play a significant role in the heterogeneous ice formation in cirrus cloud (Wise et al., 2010; Piedehierro et al., 2021)."**

*The final conclusions (e.g., lines 507-510) should be clarified and made stronger. Do you recommend using partitioning models (or two of the models; "either" in line 509)? What kind of experimental data do you need? There is at least experimental $SS_c$ data available (e.g., Abbatt et al., Atmospheric Environment, 39, 4767-4778, 2005) and direct observations at least for Malonic Acid (Blower et al., The Journal of Physical Chemistry A, 117, 2529-2542, 2013).*

A range of previous studies has shown that bulk–surface partitioning models are needed for thermodynamically consistent predictions of cloud droplet activation. Here we have shown that models currently available in literature do not always predict the same properties of activating droplets for the same aerosol systems and under the same conditions. More effort should therefore be put into validating the robustness of each of the partitioning models across a much wider range of conditions than those for which the models have so far been applied and separate evaluated against data.

A variety of experimental input data is needed to run the different models, including the liquid and solid phase properties of pure compounds, composition dependent mixture properties, and measured points along the droplet Köhler curves. Experimental $SS_c$ data for various aerosol compositions and sizes is useful to validate the model predictions of cloud droplet activation. We have included additional comparisons to experimental $SS_c$ data, also including to data from Abbatt et al. (2005), to the results of the present work. However, the comparison can only validate the predicted critical supersaturation ($SS_c$) values, not the underlying extent and representation of organic bulk–surface partitioning. Furthermore, size and composition resolved experimental $SS_c$ data is generally not available for atmospheric aerosol samples and conditions, which is among the main reasons for presenting and evaluating predictive models.

A separate paragraph has been added for the final conclusions. In addition, we clarify and expand:

**"Overall, this comparison of different model predictions for surface active aerosols clearly highlights the need for further experimental validation of the different bulk–surface partitioning models across a wide range of particle mixtures and conditions, before any of the models is used as basis for broad generalizations of results to atmospheric processes. Our results highlight how the different models may predict similar activation properties but very different degrees of organic bulk–surface partitioning in droplets. Comparison to experimental $SS_c$ values is valuable for confirming critical droplet properties predicted with the different models for a range of conditions, and for surfactants of vary-**

**ing strengths, but will not offer insight into the exact role and dynamics of bulk–surface partitioning in the droplet**
370 **activation process. Recently, Bzdek et al. (2020) presented the first direct experimental observation of the impact of size-dependent bulk–surface partitioning in droplets suspended in air. Such studies however remain highly elaborate and sparse, and therefore creating robust models and comparing different models across a variety of different systems can offer insight into the process and guide future experimental studies. In this work, we predict very similar $SS_c$ values using the Gibbs, monolayer, and compressed film partitioning models and the bulk solution model, which are**
375 **furthermore similar to the experimental $SS_c$ values, despite significant differences in predicted organic bulk–surface partitioning between the models. Given that the bulk solution model does not represent demonstrated effects of organic bulk–surface partitioning in small droplets and the compressed film model can predict unphysical droplet surface tensions, this suggests that the Gibbs and monolayer partitioning models are currently the preferable options for modeling droplet growth and activation of aerosol systems comprising moderately strong surface active compounds. A similar**
380 **comparison between the different droplet models for more strongly surface active particle components is the topic of future work. For such aerosol systems, conclusions regarding the different models may differ significantly from those of the present work."**

**1.3 Technical corrections**

385 *Below are few quotes from the text that seem to have something wrong or could just be improved for clarity.*

*Line 26: "and inducing concentration gradients"*

*Line 27: "compartments"*

Lines 26-27: "In liquid aerosol mixtures, such as aqueous droplets, surfactants can adsorb at the interfaces, lowering the surface tension and inducing concentration gradients between the droplet surface and bulk compartments."

390 We rephrase the sentence, and move clarification of the word "partitioning" to this point of the Introduction:

**"In liquid aerosol mixtures, such as aqueous droplets, surfactants can adsorb at the interfaces, lowering the surface tension and distributing their mass between the droplet bulk and surface phases. The distribution of surfactant mass between the surface and bulk of a solution is here referred to as the bulk–surface *partitioning*."**

*Lines 50-51: "They presented . . . uptake (Sorjamaa et al., 2004)."*

395 Line 50-51: "They presented thermodynamic predictions showing that partitioning of the surfactant from the droplet bulk to the surface can limit the amount of dissolved solute and, as a result, reduce hygroscopic water uptake (Sorjamaa et al., 2004). "

We rephrase the sentence to:

**"Sorjamaa et al. (2004) presented thermodynamic predictions showing that partitioning of the surfactant from the droplet bulk to the surface can limit the amount of dissolved solute and therefore reduce hygroscopic water uptake."**

400 *Line 56: "and systematically across their"*

Line 56:

"This was later demonstrated both experimentally and in thermodynamic model calculations for droplets comprising a series of atmospherically relevant fatty acid salts and systematically across their mixtures with sodium chloride (Prisle et al., 2008, 2010)."

405 We rephrase and add references at the request of referee 2:

**"This has later been demonstrated both experimentally and in thermodynamic model calculations for particles comprising a range of surface active compounds and their mixtures with soluble components (Li et al., 1998; Sorjamaa et al., 2004; Prisle et al., 2008, 2010; Kristensen et al., 2014; Hansen et al., 2015; Petters and Petters, 2016; Lin et al., 2018; Forestieri et al., 2018; Prisle et al., 2019; Prisle, 2021)."**

410 *Line 79: "aerosol: Enhancement of"*

We rephrase to:

**"aerosol. Enhancement of"**

*Line 85: "by Sorjamaa et al. (2004); Prisle et al. (2008, 2010),"*

Edited to

415 **"Sorjamaa et al. (2004), Prisle et al. (2008, 2010)**"

*Line 96: "natrium"*

Changed to the correct word: "sodium".

*Line 105: "their immediate atmospheric"*

Line 105:

420 "Moderate organic surfactants are represented by dicarboxylic acids due to their immediate atmospheric relevance (e.g. Shulman et al., 1996; Hori et al., 2003) and abundance (e.g. Khwaja, 1995; Mochida et al., 2007; Jung et al., 2010)."

We delete the unnecessary word and clarify the terminology related to the term "moderate surfactant" :

**"Surface active organics of moderate strength are represented by dicarboxylic acids due to their atmospheric relevance (e.g. Shulman et al., 1996; Hori et al., 2003) and abundance (e.g. Khwaja, 1995; Mochida et al., 2007; Jung et al.,**
425 **2010)."**

The term "moderate surfactant" has been clarified in all instances for the revised manuscript.

*Line 109: "to be about"*

"To reduce the surface tension of water in an aqueous bulk solution by 10 % at 298.15 K, the mole fraction of malonic, succinic and glutaric acids in the solution estimated from a fit (Hyvärinen et al., 2006) to be about 0.061, 0.017 and 0.0070 respectively."

430 We rephrase to:

**"To reduce the surface tension of an aqueous bulk solution at 298.15 K by 10 % from the value of pure water, the mole fraction of malonic, succinic and glutaric acids in the solution must be 0.061, 0.017 and 0.0070 respectively (Hyvärinen et al., 2006)."**

*Line 118: "Theory & Methods"*

435 We rephrase to:

**"Theory and modeling"**

*Line 141: "particle, with the"*

Line 141:

"For each dry particle size, the total amounts of ammonium sulphate salt and organic molecules are calculated based on their pure solid phase densities and relative mass fractions in the particle, with the assumption of spherical dry particles and additive solid phase volumes in all cases."

We rephrase to:

**"The calculations with all models assume spherical dry particles and additive solid phase volumes. For each dry particle size, the total amounts of ammonium sulphate and organic molecules are calculated based on their pure solid phase densities and relative mass fractions in the particle."**

*Lines 147 and elsewhere: "supplement section"*

Each such instance has been changed to the form "section SX of the supplement", X denoting the specific sections of the supplement being referred to.

*Line 165: "whence"*

We rephrase "whence" to "and therefore":

**"Compounds adsorbed at the surface are assumed to not contribute to the total droplet volume, and therefore a positive surface volume of one compound (surfactant) must be balanced by depletion of other compounds (water and salt) from the surface. "**

*Line 174: "inside"*

The word "inside" has been changed to "of all components in".

**"In addition, we assume volume additivity (such that the droplet diameter is given by the sum of individual pure component molar volumes) and mass conservation ($n_j^T = n_j^S + n_j^B$) of all components in the droplet (Prisle, 2006)."**

*Line 189: "the very surface area-to-bulk"*

The missing word "large" has been added:

**"the very large surface area-to-bulk"**

*Line 248: "For phase the bulk phase"*

We removed the extra "phase":

**"For the bulk phase"**

*Line 276: "be even distributed"*

Fixed to:

**"be evenly distributed"**

*Table 1: "'when they differ from the pure surfactant"*

Table 1 caption: "Methods of calculating the total amount of water ($n_w^T$), the water activity ($a_w$), the surface tension ($\sigma$) and the composition of droplet surface and bulk, when they differ from the pure surfactant, for the different models. Details of the activity and surface tension equations are given in the supplement."

The caption has been clarified to:

**"Methods of calculating the total amount of water ($n_w^T$) in the droplets, the droplet water activity ($a_w$) and surface tension ($\sigma$), and the composition of the droplet surface and bulk phases used with the different models. Details of the water activity and surface tension equations are given in the sections S2 and S4 of the supplement."**

*Line 312 (also 328 and 338): "0.19% supersaturation"*

The expressions all include the superfluous word "supersaturation" which have been removed. Additional clarification in notation regarding the ranges of the $SS_c$ and $d_c$ values predicted with the different models has also been added. Line 312:

**"The range of predicted $SS_c$ values increases compared to Fig. 1(a) and now $\Delta SS_c = 0.19$ %, while the predicted $d_c$ values are all within a range of $\Delta d_c = 84$ nm."**

Line 328:

**"The range of predicted $SS_c$ increases to $\Delta SS_c = 0.5$ % and the predicted $d_c$ range also increases to $\Delta d_c = 199$ nm."**
Line 338:

**"The increasing trend in the width of the ranges of predicted $SS_c$ and $d_c$ with organic mass fraction in the dry particle continues to $\Delta SS_c = 1.31$ % and $\Delta d_c = 282$ nm, respectively.**

*Line 243: "predictions will bias SSc values too high"*

" Even though the experimental data is obtained at slightly different conditions than the present simulations, it is reasonable to assume that the latter model predictions will bias $SS_c$ values too high, as underlying assumptions may no longer be representative for the highest organic fractions in the particles."
We have clarified the expression:

**"Although the experimental data is obtained at slightly different conditions than the present simulations, comparison with the $SS_c$ prediction of the simple model indicates that the model predicts $SS_c$ to be too high because underlying assumptions of complete surface partitioning are no longer representative for the highest organic fractions in the particles."**

*Line 377: "The pure surface tension of malonic acid"*

"The pure surface tension of malonic acid is estimated according to Hyvärinen et al. (2006) to be about 48.24 $\mathrm{mN\,m^{-1}}$ at 298.15 K, which is lower than any surface tension depression reported by Hyvärinen et al. (2006)."
We rephrase to:

**"The surface tension of pure supercooled malonic acid is estimated by Hyvärinen et al. (2006) as 48.24 $\mathrm{mN\,m^{-1}}$ at 298.15 K, which is lower than any surface tension reported by the same study for the aqueous solutions of the organic acids."**

*Line 388-389: "fraction of 0.965 the ammonium"*

"This was also found for each of the organic acids in this study, except at the highest mass fraction $w_{\mathrm{p,org}} = 0.95$, which would correspond to a suberic acid volume fraction of 0.965 the ammonium sulphate particles of Davies et al. (2019)."
We clarify (and recalculate) to:

**"This was also found for each of the organic acids in the present work, except at the highest mass fraction $w_{\mathrm{p,org}} = 0.95$, which would correspond to a suberic acid volume fraction of 0.964 in particles mixed with ammonium sulphate."**

*Line 405: "(Eq. (S9) in"*

"This indicates that the droplet compositions are similar with all three models, as surface tension has been evaluated from the same composition-dependent function (Eq. (S9) in the supplement)."

Clarification has been added:

**"This indicates that the predicted droplet compositions are similar for all three models, as the droplet surface tension has been evaluated from the same composition-dependent function (given in Eq. (S9) of the supplement)."**

*Line 458: "(Eq. (6))."*

"The minimum value is due to the unconstrained partitioning factor and has no physical meaning for the Köhler curve (Fig. 1) until the surface tension (Fig. 2) starts increasing from its minimum value (Eq. (6)). This is because the surface tension is the quantity being constrained by the parameter $\sigma_{\min}$ in the framework."

Clarification has been added:

**"The minimum value of the curve is due to the unconstrained partitioning factor that does not affect the Köhler curve (Fig. 2) until the surface tension (Fig. 3) starts increasing from its minimum value. The surface tension is being constrained by the model parameter $\sigma_{\min}$, and the fraction of the molecules partitioned to the surface ($f_{\mathrm{surf}}$) has no effect on the EoS while the constraint applies (see Eq. (3)- (6))."**

*Line 469: "approximately 0.65."*

"With the compressed film model, the predicted partitioning factors are higher, but comparable to the results of Ruehl et al. (2016), where the values for malonic acid dry particles of 150 nm and $w_{\mathrm{p,org}} = 0.96$ are always larger than approximately 0.65."

The sentence has been changed to: **"The predicted partitioning factors with the compressed film model are higher, but comparable to the results of Ruehl et al. (2016), where the values for malonic acid dry particles of 150 nm and $w_{\mathrm{p,org}} = 0.96$ are always above a minimum of 0.65 (estimated by visual inspection of Fig. S4 from the supplement of Ruehl et al. (2016))."**

*References: please reorder references with the same first author according to the ACP standards*

The Copernicus LaTeXtemplate automatically assigns the order of the references.

**2  Referee 2**

*Vepsäläinen, Calderón, Malila, and Prisle contribute a comparison of simplified formulations for treating surface tension in droplet nucleation (manuscript #acp-2021-561). They compare six published thermodynamically consistent models of the bulk-to-surface partitioning behaviour of moderately strong surfactants and the effects this has on droplet surface tension, water uptake, and cloud droplet nucleation. For mixtures of dicarboxylic acids with ammonium sulphate, cloud droplet activation predictions were similar for lower organic fractions (organic < 50% of dry aerosol mass), but for dry organic fractions greater than 80%, differences emerge based on the partitioning calculations.*

*Predicting cloud droplet nucleation from aerosol chemistry is challenging and of interest to the broader atmospheric chem-*

540 *istry community. Though the models are all previously published, the direct comparison between the models is useful. The models ingest parameters derived from experimental data, but the present paper does not include a comparison of the model to measurements. I recommend publication if the authors can address several shortcomings of the present manuscript.*

**2.1 Major comments**

545 *Sensitivity analysis is provided for the compressed film model. However, for all models, various parameters are taken from measurements and could vary as a function of composition, temperature, and mixing state (among others). The manuscript presents the implementation of six published models with no additional experiments nor a comparison to published measurements. The value of the publication is in the direct comparison between formulations. The community gains nothing from this paper if the results do not include analysis of the sensitivity of each model to perturbation in the input parameters.*

550 Predictive models can be used to obtain droplet properties where measurements are not possible and therefore comparing state-of-the-art models in terms of their ability to predict similar properties for common aerosol systems has significant value in itself and is the main motivation of the current work. This point has also been addressed earlier in this response in relation to one of the first comments from referee 1 (Section 1.2). Comparison to a single experimental critical supersaturation value was made in Fig. 1(d) of the original manuscript and we have included additional experimental critical supersaturation values 555 in the revised manuscript.

A comprehensive sensitivity analysis of all models with respect to input parameters has been added to the revised supplement. More details on the reasoning for the original sensitivity analysis are given in response to the comment from referee 1 about Section 3.1.2 of the original manuscript.

*None of the models treat nonideal interactions of the organic with the other solution components. This is a significant* 560 *simplifying assumption that should be discussed. Especially at the onset of water uptake and early droplet growth, nonideal solution activity influences the Köhler curve. In the supplement, after equation S7 the activity is expressed $a_w = x_w \gamma_w$. The activity coefficient is not defined or used and is subsumed into the constants of equation S7. This is the only activity coefficient in the paper, and it appears not to have been used in the calculations. Prediction of activity, independent of surface partitioning, is also an active area of research, particularly for organic/inorganic mixtures. How would nonideal activity introduce error in* 565 *the authors' calculations? This needs to be discussed.*

The sentence after Eq. S7 defining calculations as $a_w = x_w \gamma_w$ was a mistake and has been deleted in the revised manuscript. It was accidentally left in after unifying the presentation of the all the equations for calculating $a_w$ just before the initial submission of the original manuscript.

Here we present a comparison of state-of-the art bulk–surface partitioning models currently available to the community. 570 We fully agree that not treating non-ideal interactions of the organic with the other solution components is a limitation of the models in the form they have been presented. A significant challenge is that activity coefficients and other parameters needed for a thermodynamically consistent account of non-ideal water activities are not available for aqueous mixtures of compositions corresponding to many solution states realized by growing and activating droplets. Most previous works have assumed ideal activities for some or all droplet components in various Köhler calculations and it is outside the scope of this work to amend these challenges, which have already persisted for decades. A formal description of nonidealities in aqueous solutions of dicarboxylic acids and inorganic salts is available elsewhere (Clegg and Seinfeld, 2006a, b; Tong et al., 2008). Prisle et al. (2010) presented a sensitivity analysis of predictions of droplet activation using a bulk–surface partitioning model with respect to the activity of water. Prisle (2021) used a fully ternary experimental $a_w$ parameterization to capture all effects of non-ideal water activity in similar predictions. Both studies concluded that for particles comprising strong surfactants, predictions of droplet activation are not very sensitive to water non-ideality for most droplet states, because droplets are highly dilute due to bulk–surface partitioning. This effect is also seen in the present work, albeit less pronounced due to the moderate strength of the investigated surfactants. We are combining the changes made to the manuscript here with a comment from Section 1.2 of the response by referee 1. We have added the following clarification regarding activities to section 2.1 of the manuscript:

"The water activity of the binary mixture of ammonium sulphate and water used with the simple model is calculated using a parametrization (Prisle, 2006) that has been previously used together with the model (e.g. Prisle et al., 2019; Prisle, 2021). Similarly to the original work describing the partial organic film model (Ovadnevaite et al., 2017), the activity used for this model is based on AIOMFAC calculations (Zuend et al., 2008, 2011; AIOMFAC-web). A significant challenge for Köhler calculations is that activity coefficients are not typically available for all droplet components and at solution states corresponding to growing and activating droplets. For some mixtures, activities can vary significantly across the relevant range of droplet compositions (e.g. Hyttinen et al., 2020; Michailoudi et al., 2020). Robust composition dependent activity relations are exceedingly difficult to obtain due to the challenges of related to their direct measurements for non-volatile and trace components and as activities cannot be inferred from the Gibbs–Duhem equation for higher order mixtures. To our knowledge, experimental data is not available to obtain a complete description of the non-ideal interactions in the ternary organic-inorganic aqueous mixtures relevant for the present work. Therefore, in the cases where the droplet bulk is a ternary mixture (Gibbs, monolayer, compressed film and bulk solution models), we calculate $a_w$ as a corrected molar fraction including non-ideal effects of ammonium sulphate on water (see Eq. (S5) of the supplement). Non-ideal effects can be included implicitly by using composition dependent experimental data in cases where such data is available. The monolayer model (Malila and Prisle, 2018) implicitly includes non-ideal solution interactions through composition dependent experimental solution surface tension and density. However, for the droplet compositions in the present work, densities of ternary solutions are calculated for pseudo–binary ideal mixtures of water-salt and organic (section S4.2 of the supplement) and therefore do not fully capture all non-ideal interactions. Prisle (2021) directly used a fully ternary experimental $a_w$ parameterization together with a Gibbs bulk–surface partitioning model,which is limited to the specific aqueous mixtures of NaCl and Nordic Aquatic Fulvic Acid studied. The compressed film model (Ruehl et al., 2016) parameters are fitted to experimental Köhler growth curve values and therefore includes an averaged account of the non-ideal solution interactions across the droplet states spanned by these experiments."

In addition, section S2.3 of the supplement has been modified to include the following text regarding Eq. S5:

**"Frosch et al. (2011) found that water activities of organic-inorganic aqueous mixtures similar to the droplets of the present work were mainly influenced by the presence of inorganic salts due to the higher degree of dissociation of the salt. Therefore, we here assume van't Hoff factors of unity for each of the organics and for ammonium sulphate, the van't Hoff factor is calculated as..."**

*It would improve the impact of the paper if a schematic illustration were introduced to describe and differentiate between the models, given that the present manuscript presents the implementation of published equations.*

A conceptual figure illustrating the initial dry particles at a range of compositions, and the different models used to predict droplet critical properties at some point before droplet activation has been added as Fig. 1 of the revised manuscript.

**2.2  Technical**

*Line 5-6: you might rephrase to clarify and list the surface-active materials*

The surface active compounds have been named in the abstract:

**"In the current work, we have used six different approaches documented in the literature to represent surface activity in Köhler calculations of cloud droplet activation for particles consisting of one of three moderately surface active organics (malonic, succinic or glutaric acid) mixed with ammonium sulphate in different mass ratios."**

*Line 41: can you clarify "microscopic and submicron"*

"These works did not consider that surface adsorption and the resulting concentration gradient between surface and bulk phases can also lead to significant depletion of the bulk phase in microscopic and submicron droplets typically involved in cloud droplet activation, due to their large surface area-to-bulk volume ratios (e.g. Prisle et al., 2010; Bzdek et al., 2020; Lin et al., 2020)."

Clarification has been added:

**"These works did not consider that surface adsorption and the resulting concentration gradient between surface and bulk phases can also lead to significant depletion of the bulk phase in microscopic and submicron droplets (i.e. with diameters in the micrometer range or smaller, or below 1 $\mu$m, respectively) typically involved in cloud droplet activation, due to their large surface area-to-bulk volume ratios (e.g. Prisle et al., 2010; Bzdek et al., 2020; Lin et al., 2020)."**

*Line 57: you might consider citing all papers that have demonstrated this, including more recent papers and those from other groups.*

Assuming this refers to:

"...the isolated consideration of surface tension depression in cloud droplet activation without accounting for the surfactant partitioning effect on bulk hygroscopicity can lead to exaggerated potential of cloud droplet nuclei (CCN) activation. This was later demonstrated both experimentally and in thermodynamic model calculations for droplets comprising a series of atmospherically relevant fatty acid salts and systematically across their mixtures with sodium chloride (Prisle et al., 2008, 2010)."

References have been added and the sentence has been reformulated:

**"This has later been demonstrated both experimentally and in thermodynamic model calculations for particles comprising a range of surface active compounds and their mixtures with soluble components (Li et al., 1998; Sorjamaa**

et al., 2004; Prisle et al., 2008, 2010; Kristensen et al., 2014; Hansen et al., 2015; Petters and Petters, 2016; Lin et al., 2018; Forestieri et al., 2018; Prisle et al., 2019; Prisle, 2021)"

*59: "finite-sized" please give the size range*

The sentence has been changed to include the specific range from the study referenced:

**"The predicted bulk–surface partitioning effects were only recently verified experimentally for finite-sized droplets (specifically of 7-9 $\mu$m radius, but more generally referring to microscopic particles and droplets with finite surface-area-to-bulk-volume ratios) suspended in the air (Bzdek et al., 2020)."**

*71-74: this seems redundant*

Lines 69-75:

"Fatty acids and their salts are a major class of organic compounds identified in atmospheric aerosols (e.g. Mochida et al., 2002, 2007; Cheng et al., 2004; Li and Yu, 2005; Forestieri et al., 2018) and often are relatively strong surfactants (see e.g. Prisle et al. (2008, 2010) and references therein), but also surfactants of moderate strength are abundantly present in the atmosphere. In microscopic droplets, strong surfactants are predicted to be strongly depleted from the bulk phase due to bulk–surface partitioning, leading to modest effective hygroscopicity of the surface active component, while at the same time, the finite amount of surface active material is still not sufficient to significantly reduce droplet surface tension (Prisle et al., 2008, 2010, 2011; Lin et al., 2020; Bzdek et al., 2020)."

The section has been edited and shortened:

**"... but also surfactants of moderate strength are abundantly present in the atmosphere. The behaviour observed for the fatty acid salts may however be a limiting case..."**

*96: before, this was called sodium. Why natrium now?*

The word has been changed to "sodium" for consistency.

*248: "for phase the bulk phase " a typo?*

The extra "phase" has been removed.

*Please define and quantify the term, 'moderate surfactant'*

For the revised version of the manuscript, we have removed the term "moderate surfactant". The term was intended to mean surface active compounds of moderate strength. Following the definition given in the original manuscript starting at line 109: "We quantify surfactant strength similarly to Prisle et al. (2010, 2011) as the surface tension reduction from that of pure water at a given surfactant bulk phase concentration. To reduce the surface tension of water in an aqueous bulk solution by 10 % at 298.15 K, the mole fraction of malonic, succinic and glutaric acids in the solution needs to be about 0.061, 0.017 and 0.0070 respectively estimated from a fit (Hyvärinen et al., 2006). In addition, the dicarboxylic acids used for this study can reduce surface tension to roughly 50 mN m$^{-1}$ at most in large enough concentrations (Hyvärinen et al., 2006; Booth et al., 2009). Stronger surfactants, such as fatty acid salts, can reduce surface tension to 20–30 mN m$^{-1}$ at most (Prisle et al., 2010)."

This is a qualitative definition intended to provide a conceptual grouping of the compounds that are the focus of this work. In the revised manuscript, the description has been moved from the introduction to Section 2 and edited in the process:

**"We describe surfactant strength similarly to Prisle et al. (2010, 2011) in terms of the ability to reduce the surface tension**

**from that of pure water at a given surfactant bulk phase concentration. To reduce the surface tension of an aqueous bulk solution at 298.15 K by 10 % from the value of pure water, the mole fraction of malonic, succinic and glutaric acids in the solution must be 0.061, 0.017 and 0.0070 respectively (Hyvärinen et al., 2006). Furthermore, in sufficiently large concentrations the dicarboxylic acids studied here can at most reduce aqueous surface tension to roughly 50 $\mathrm{mN\ m^{-1}}$ (Hyvärinen et al., 2006; Booth et al., 2009). Stronger surfactants, such as fatty acid salts, can reduce surface tension to 20–30 $\mathrm{mN\ m^{-1}}$ and a given surface tension reduction occurs at much lower aqueous concentrations (Prisle et al., 2010)."**

*Line 397 - Pure malonic acid is a powder; is this surface tension the CMC value?*

The surface tension value is that of supercooled liquid malonic acid calculated according to the fit of Hyvärinen et al. (2006). Clarification has been added into several locations of section 3.2.

*394,395 – surface tension is reduced by higher concentration, or as a size-dependent effect?*

Lines 394-395: "For each of the models, predicted surface tension can be significantly reduced at the smaller droplet sizes in the beginning of the growth curve where droplets are most concentrated. However, in most cases, the surface tension at the point of activation is close to the pure water value."

The composition of the dry particles is defined as an input of the models. This defines the absolute amounts of salt and surfactant solute in the growing droplets. The concentration of surfactant in the droplet at smaller sizes early on the Köhler growth curve is higher, as the absolute amount of salt and organic in the droplets does not change, only the amount of water.

Line 394-395 has been modified as:

**"For each of the models, predicted surface tension can be significantly reduced at the smaller droplet sizes at the beginning of the growth curve where the surfactant is most concentrated in the droplets."**